# Monoacylglycerol lipase regulates cannabinoid receptor 2-dependent macrophage activation and cancer progression

Wei Xiang[1], Rongchen Shi[1], Xia Kang[1], Xuan Zhang[2], Peng Chen[3], Lili Zhang[1], Along Hou[1], Rui Wang[1], Yuanyin Zhao[1], Kun Zhao[1], Yingzhe Liu[1], Yue Ma[1], Huan Luo[1], Shenglan Shang[1], Jinyu Zhang[4], Fengtian He[1], Songtao Yu[2], Lixia Gan[1], Chunmeng Shi[5], Yongsheng Li [6], Wei Yang [7], Houjie Liang[2] & Hongming Miao[1]

Metabolic reprogramming greatly contributes to the regulation of macrophage activation. However, the mechanism of lipid accumulation and the corresponding function in tumor-associated macrophages (TAMs) remain unclear. With primary investigation in colon cancer and confirmation in other cancer models, here we determine that deficiency of mono-acylglycerol lipase (MGLL) results in lipid overload in TAMs. Functionally, macrophage MGLL inhibits CB2 cannabinoid receptor-dependent tumor progression in inoculated and genetic cancer models. Mechanistically, MGLL deficiency promotes CB2/TLR4-dependent macro-phage activation, which further suppresses the function of tumor-associated CD8+ T cells. Treatment with CB2 antagonists delays tumor progression in inoculated and genetic cancer models. Finally, we verify that expression of macrophage MGLL is decreased in cancer tissues and positively correlated with the survival of cancer patients. Taken together, our findings identify MGLL as a switch for CB2/TLR4-dependent macrophage activation and provide potential targets for cancer therapy.

[1] Department of Biochemistry and Molecular Biology, Third Military Medical University, Chongqing 400038, China. [2] Department of Oncology, Southwest Hospital, Third Military Medical University, Chongqing 400038, China. [3] Department of General Surgery, PLA 324 Hospital, Chongqing 400020, China. [4] National Engineering Research Center of Immunological Products, Third Military Medical University, Chongqing 400038, China. [5] Institute of Combined Injury, State Key Laboratory of Trauma, Burns and Combined Injury, Third Military Medical University, Chongqing 400038, China. [6] Clinical Medicine Research Center & Institute of Cancer, Xinqiao Hospital, Third Military Medical University, Chongqing 400037, China. [7] Department of Pathology, School of Basic Medical Sciences & Nanfang Hospital, Southern Medical University, Guangzhou 510515, China. These authors contributed equally: Wei Xiang, Rongchen Shi, Xia Kang, Xuan Zhang. Correspondence and requests for materials should be addressed to Y.L. (email: yli@tmmu.edu.cn) or to W.Y. (email: yangwei@sibcb.ac.cn) or to H.L. (email: lianghoujie@sina.com) or to H.M. (email: hongmingmiao@sina.com)

Conventional cancer therapies, including surgery, cytotoxic chemotherapy, and radiation, aim to eradicate malignant cells. However, cancer cells do not grow in isolation, and stromal cells (T cells, macrophages, etc.) in the tumor microenvironment also need be targeted for effective therapeutic outcomes[1–3]. The outcomes can be achieved directly via the main effectors of the immune system, cytotoxic CD8+ T cells (as are used in checkpoint blockade strategies), or indirectly via targeting other immune cell types, such as macrophages[3]. Blockade of CSF-1/CSF-1R signaling depletes macrophages and stimulates CD8+ T cell responses, resulting in decreased tumor progression in mouse models of breast and cervical cancers[4].

Despite of the complex phenotypes (heterogeneity) in vivo[5], macrophages are ideally defined as two extremes in vitro: "classically activated" (or "M1") macrophages or "alternatively activated" (or "M2") macrophages[6]. M1 macrophages are polarized in settings of local interferon gamma (IFNγ)-producing Th1 responses, whereas M2 macrophages respond to cytokines characteristic of Th2 responses, such as IL-4 and IL-13. Notably, tumor-associated macrophages (TAMs) are prone to M2-like phenotypes, producing Th2 cytokines and subsequently promoting tumor progression[7]. However, how the TAMs are re-educated to the M2-like phenotype is still not clear.

Emerging evidence has revealed that metabolic reprogramming greatly contributes to the regulation of macrophage activation. In lipid metabolism, saturated free fatty acids induce proinflammatory activation via toll like receptor 4 (TLR4) and subsequent NF-κB as well as JNK pathways[8]; in our previous work, we revealed that deficiency of AB-hydrolase containing 5 (ABHD5), a coactivator of adipose triglyceride lipase[9], stimulated NLRP3-inflammasome-dependent proinflammatory activation[10]. The tumor microenvironment is a special niche characterized by ischemia, hypoxia, acidity, and innutrition[11]. All the stromal cells likely undergo a special metabolic reprogramming to adapt and survive in this environment. We and others have reported that lipid metabolism in cancer-associated myeloid cells is largely altered[2, 12]. However, how lipids were accumulated and the corresponding function in TAMs remains unclear.

In the present study, we screened lipid metabolism-related genes in TAMs and found that deficiency of monoacylglycerol lipase (MGLL) contributed to lipid accumulation, macrophage activation, CD8+ T cell inhibition and tumor progression in inoculated and genetic cancer models. We also explored the mechanism underlying MGLL-CB2-regulated macrophage activation using in vitro and mouse models with pharmacological or genetic manipulation. Our findings indicate that modulation of MGLL-CB2 axis in macrophages could be a promising strategy for cancer treatment.

## Results

### MGLL deficiency in TAMs contributes to lipid accumulation.
We set up a variety of subcutaneous tumor models to observe the roles of TAMs in tumor progression by employing two colorectal cancer cell lines (CT-26 and MC-38) and a breast cancer cell line 4T1. We demonstrated that the numbers of TAMs increased over time in the CT-26, MC-38, and 4T1 tumor models (Fig.1a, b). With Bodipy staining, we revealed that the TAMs from MC-38 tumor contained notably more lipids than the spleen macrophages from the corresponding tumor-bearing mice (Fig. 1c). This finding was confirmed by flow cytometry analysis in CT-26, MC-38, and 4T1 tumor models (Fig. 1d and Supplementary Fig. 1a). These experiments indicated that macrophages accumulated lipids in tumor environments.

To explore the potential mechanism involved in lipid accumulation, we analyzed the lipid metabolism-related gene expression in CT-26 cell-stimulated macrophages according to our previous gene-microarray analysis[2]. As shown in Fig. 1e, ABHD5, a lipolytic factor of triglycerides, was increased, whereas MGLL, the key lipase of monoacylglycerols was downregulated in CT-26-stimulated macrophages. Therefore, we presumed that MGLL deficiency might be a key factor contributing to lipid deposition in TAMs. To verify this presumption, we isolated macrophages from spleens or tumor tissues of different tumor models. We demonstrated that the expression of MGLL, but not other lipases, such as ATGL or HSL, was largely decreased in TAMs (Fig. 1f and Supplementary Fig. 1b). Consistent with our previous study[2], ABHD5 expression in TAMs was increased (Supplementary Fig. 1b). We further confirmed that CT-26, MC-38, and 4T1 cells suppressed macrophage MGLL mRNA level (Supplementary Fig. 1c) and lipolytic activity (Supplementary Fig. 1d). In line with the gene expression data, we found that the levels of triglycerides, diglycerides, and monoglycerides in TAMs notably increased (Fig. 1g). In vitro, we also demonstrated that MC-38 cells induced lipid accumulation in Raw264.7 cells, and this effect was prevented by constitutive expression of MGLL protein (Supplementary Fig.1e-g). To further verify the role of MGLL in lipid accumulation in TAMs, we constructed a transgenic mouse model with specific overexpression of MGLL (Tg^MGLL) in myeloid cells (Supplementary Fig. 1h-j). We demonstrated that MC-38 tumor-suppressed MGLL mRNA level could be fully rescued by macrophage Tg^MGLL (Fig. 1h). Consequently, MC-38 tumor-induced lipid accumulation in TAMs was completely prevented (Fig. 1i). Thus, the aforementioned results indicated that cancer cell-induced MGLL deficiency resulted in lipid accumulation in TAMs.

### Macrophage MGLL inhibits tumor progression.
To explore the role of macrophage MGLL in tumor progression, we set up injected tumor models in the WT and Tg^MGLL mice. In subcutaneous tumor models, the inoculated MC-38 tumors in Tg^MGLL mice grew much slower than those in the WT groups (Fig. 2a). Consistently, the transgene of MGLL in macrophages largely prolonged the survival of MC-38 tumor-bearing mice (Fig. 2b). To confirm the role of macrophage MGLL in tumor progression, we set up a conditional knockout mouse model with MGLL deficiency in myeloid cells (Myeloid-mgll-KO) (Supplementary Fig. 2a, b). In subcutaneous tumor models, Myeloid-mgll-KO markedly promoted the tumor growth and shortened the survival of MC-38 tumor-bearing mice (Fig. 2c, d).

In intravenous tumor models, the incidence of lung metastasis in MC-38 tumors was 100%, regardless of WT or Tg^MGLL mice (Fig. 2e). However, the incidence of liver metastasis was largely reduced in Tg^MGLL mice versus the WT group (Fig. 2e). In addition, we further observed the tumor lesions in lung metastasis. In comparison with WT mice, the Tg^MGLL group had dramatically decreased tumor lesions in the lungs (Fig. 2f-h). Unexpectedly, compared with the WT group, the conditioned medium of Tg^MGLL macrophages did not affect the growth and migration activity of MC-38 cells in vitro (Supplementary Fig. 2c-e). These experiments indicated that MGLL-regulated macrophage secreta did not affect tumor cell biology directly.

### Macrophage MGLL inhibits tumor progression via CD8+ T cells.
Macrophage-induced immunosuppression is suggested to be a major contributor to tumor progression[13, 14]. To observe whether macrophage MGLL would affect tumor development via adaptive immunity, the Rag1 knockout mouse model with lymphocyte deficiency was employed. We demonstrated that the anti-tumor effect of macrophage MGLL was largely attenuated in Rag1 knockout mouse (Fig. 2i), indicating an adaptive immunity

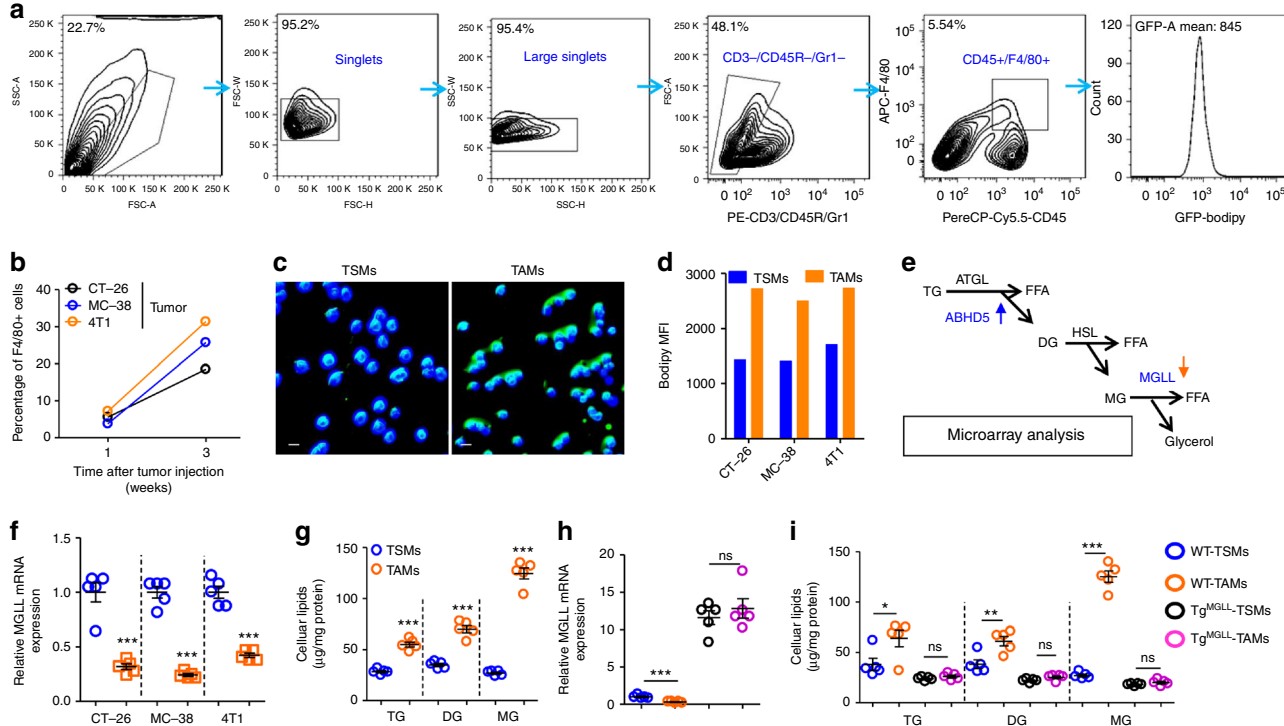

**Fig. 1** MGLL deficiency in tumor-associated macrophages results in lipid accumulation. **a** FACS gating strategy for tissue macrophages and lipid measurement. Debris and doublets were removed, and tissue macrophages were then assessed as CD3-CD45R-Gr1-CD45+F4/80+. The average fluorescence degree of macrophages stained by Bodipy (GFP) was measured. **b** The percentage of TAMs in inoculated tumors. Six-week-old WT mice were subcutaneously inoculated with CT-26, MC-38 or 4T1 cells and the TAMs were quantified at 1st and 3rd week. Each tested sample was pooled from five individual ones. **c** Lipid staining of macrophages from spleens (TSMs) or tumors (TAMs). Six-week-old mice were subcutaneously inoculated with MC-38 tumors and sacrificed two weeks later. Tissue macrophages were isolated and stained with Bodipy (Green). The nucleus was visualized by DAPI staining (Blue). This experiment was repeated four times. Representative images are displayed. Scale bars, 10 μm. **d** The lipid levels in TSMs and TAMs. Six-week-old mice were subcutaneously injected with indicated cells. Two weeks later, TSMs and TAMs were isolated for lipid staining with bodipy. The Geometric mean fluorescence intensity (MFI) of Bodipy in each group was measured. Each tested sample was pooled from five individual ones. **e** A diagram of glycerolipid metabolism. The blue arrow indicates up-regulation, and the orange arrow indicates down-regulation. A gene-microarray analysis was performed on the peritoneal macrophages, which were treated with regular or conditioned medium from CT-26 cells for 24 h. TG triglyceride, DG diglyceride, MG monoglyceride. **f** Relative mRNA expression of MGLL in TSMs and TAMs. Six-week-old mice were subcutaneously inoculated with indicated tumors and sacrificed two weeks later. TSMs and TAMs were isolated for real-time PCR assays. **g** Cellular lipid levels were measured in TSMs and TAMs from the MC-38 tumor-bearing mice as described in **c**. **h** Relative mRNA expression of MGLL in TSMs and TAMs. Six-week-old WT or myeloid MGLL transgenic (TgMGLL) mice were subcutaneously inoculated with MC-38 tumors for 2 weeks. TSMs and TAMs were then isolated for mRNA assays of MGLL. **i** Cellular lipid levels in TSMs and TAMs from the MC-38 tumor-bearing mice as described in **h**. Data in **f-i** represent the means ± s.e.ms ($n = 5$, *$P < 0.05$, **$P < 0.01$, ***$P < 0.005$; student's $t$-test; ns not significant)

dependent manner was involved. We next measured the activity of tumor-associated CD4+ and CD8+ T cells in MC-38 tumor models. We found that the tumor-associated CD8+ T cells from the TgMGLL mice expressed notably higher level of IFNγ than the WT mice (Fig. 2j and Supplementary Fig. 2f, g), indicating a dominate role of TgMGLL on CD8+ T cells, but not on CD4+ T cells. Consistently, the MGLL transgene-induced anti-tumor effects (tumor volume and survival time) in MC-38 tumor models were fully prevented by anti-CD8 antibodies (Fig. 2k, l).

**MGLL promotes proinflammatory cytokine expression in TAMs**. We next investigated the polarization status of TAMs, because macrophage activity was associated with adaptive immune response. We revealed that the percentage of M1-like TAMs was potentiated and the percentage of M2-like TAMs was decreased in the TgMGLL mice versus the WT group in MC-38 tumor models (Fig. 3a and Supplementary Fig. 3a). This finding indicated a potential role of MGLL in the M1/M2 switch in TAMs.

We further validated that the MC-38 tumors from the TgMGLL mice expressed more proinflammatory cytokines, such as IL-1β, TNFα and IFNγ, and less anti-inflammatory cytokines, such as IL-10, Arg-1, TGFβ and IL-4, than the tumors from WT mice (Supplementary Fig. 3b). However, cytokine expression did not differ between the WT and TgMGLL macrophages from the spleens of MC-38 tumor-bearing mice (Supplementary Fig. 3c). We further dynamically observed the cytokine expression in TAMs of WT or TgMGLL mice. As shown in Fig. 3b, the levels of proinflammatory cytokines reached peaks one or two weeks after tumor injection, whereas the levels of anti-inflammatory cytokines increased continuously. Notably, the TgMGLL group expressed obviously higher levels of proinflammatory cytokines and lower levels of anti-inflammatory cytokines in MC-38 tumor-derived TAMs (Fig. 3b).

To observe whether MGLL would switch M1/M2 polarization in macrophages, we isolated bone marrow-derived macrophages (BMDMs) and peritoneal macrophages (PMs) to test the hypothesis using a standard protocol for M1 or M2 induction[6]. We demonstrated that the BMDMs with the MGLL transgene

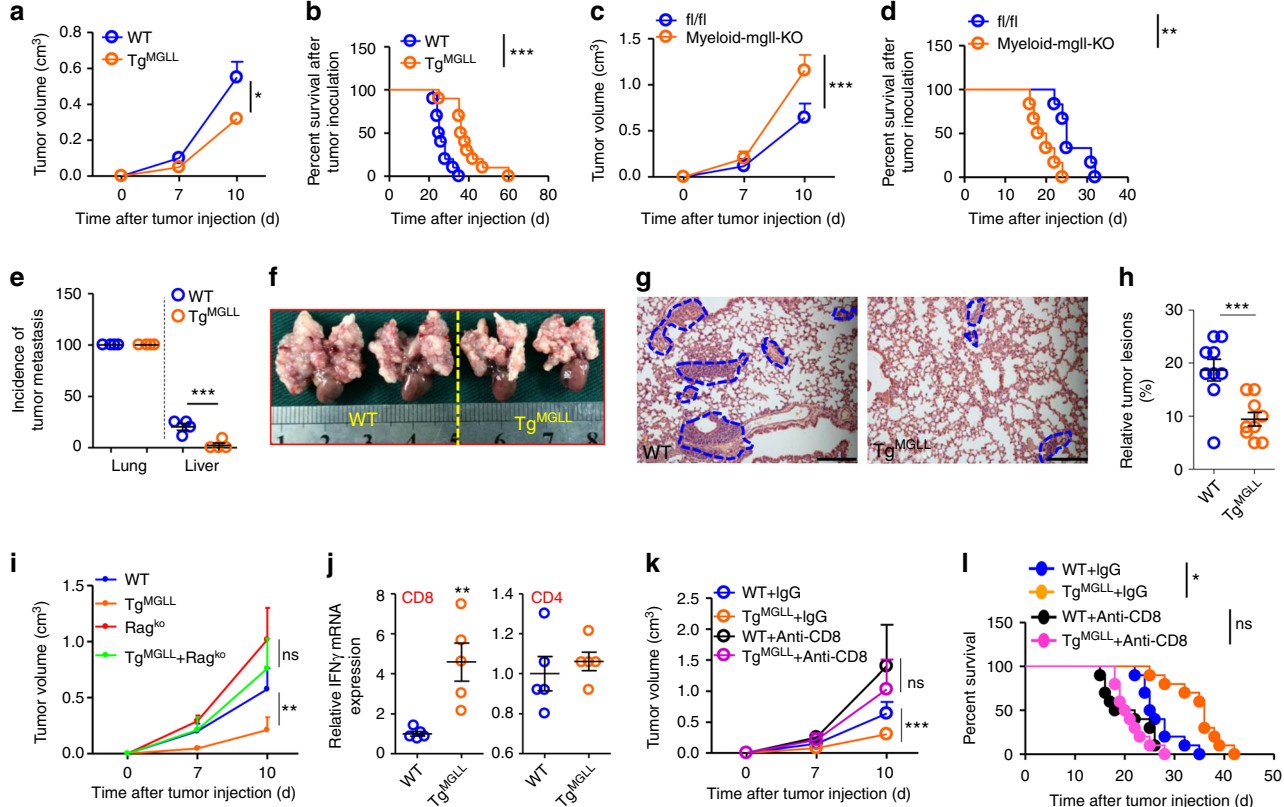

**Fig. 2** Macrophage MGLL inhibits tumor progression via CD8+T cells. **a** Six-week-old WT and Tg[MGLL] mice were subcutaneously inoculated with MC-38 cells, and tumor volumes were measured dynamically. This test was performed four times. (n = 10) **b** The survival time of the MC-38 tumor-bearing mice described in **a**. The data represent the means ± s.e.ms. (n = 10, ***P < 0.005, Gehan-Breslow-Wilcoxon test). **c** The growth curves of the subcutaneous MC-38 tumors in Myeloid-mgll-KO mice and the control littermates (fl/fl). (n = 6). **d** The survival time of the MC-38 tumor-bearing mice described in **c**. The data represent the means ± s.e.ms. (n = 6, **P < 0.01, Gehan-Breslow-Wilcoxon test). **e** Incidence of metastasis in lungs and livers of the intravenous tumor model. The 6-week-old C57BL/6 WT or transgenic mice were intravenously injected with MC-38 colorectal cancer cells (5 × 10^6 cells per mouse) via the tail vein. The mice were sacrificed 2 weeks after tumor inoculation. The lungs and livers were dissected for pathological observation of tumor lesions. Data are means ± s.e.ms. and polled from four individual experiments. Each symbol represents an experimental replicate (n = 8~11). **f–h** The MC-38 tumor lesions in lungs of WT and Tg[MGLL] mice, as described in **e**. Representative images of gross anatomy (**f**) and H&E staining (**g**) are displayed. The blue dotted lines indicate the tumor lesions. The relative tumor lesions were calculated (**h**). Scale bars, 200 μm. (n = 8~9). **i** Macrophage MGLL inhibited tumor growth in a Rag-1 dependent mannaer. MC-38 tumors were inoculated subcutaneously in WT, Tg[MGLL], Rag1[KO] or Tg[MGLL] + Rag1[KO] mice and the tumor volumes were measured dynamically. (n = 5) **j** mRNA levels of IFNγ in CD8+ or CD4+ T cells that were isolated from the MC-38 tumors inoculated subcutaneously in WT or Tg[MGLL] mice for 3 weeks. (n = 5). **k** Six-week-old WT or Tg[MGLL] mice were subcutaneously inoculated with MC-38 cells and treated with anti-CD8 antibody or IgG as a control. The tumor volume was measured dynamically. (n = 10). **l** The survival time of the MC-38 tumor-bearing mice treated as described in **k**. The data represent the means ± s.e.ms. (n = 10, *P < 0.05, Gehan-Breslow-Wilcoxon test; ns, not significant). Data in **a**, **c** and **e–k** represent the means ± s.e.ms. (*P < 0.05, **P < 0.01, ***P < 0.005; student's t-test; ns not significant)

expressed much higher levels of IL-1β and TNFα in response to IFNγ and LPS stimulation and notably lower levels of IL-10, Arg-1, and TGFβ in response to IL-4 treatment (Fig. 3c). Identical results were obtained when these experiments were performed with the PMs (Supplementary Fig. 3d). Furthermore, we also demonstrated that MGLL knockout in macrophages attenuated IL-1β and TNFα mRNA expression in response to IFNγ and LPS stimulation and potentiated the expression of IL-10, Arg-1, and TGFβ in response to IL-4 treatment in BMDMs (Fig. 3d).

**MGLL regulates macrophage activity via CB2/TLR4 interaction.** N-arachidonoylethanolamide and 2-arachidonoylglycerol (2-AG) are major ligands of CB1 and CB2 cannabinoid receptors, forming the endocannabinoid system, and play a modulating role in immune and inflammation responses[15]. The CB1 receptors are found primarily in the central nervous system[16]. By contrast, CB2 receptors predominate in the immune system[17]. It should be pointed out that 2-AG is broke down by the catabolic enzyme

MGLL[18]. Consistently, we demonstrated that the levels of 2-AG in TAMs increased dramatically and were reduced by the MGLL transgene in macrophages (Fig. 4a). Furthermore, we found that CB2 mRNA level was also stimulated in TAMs in multiple tumor models (Fig. 4b), whereas CB1 expression in TAMs was not detectable with real-time PCR assays (No CT values could be collected). Interestingly, the mRNA levels of CB2 in TAMs increased over time in MC-38 tumor-bearing mice (Fig. 4c). The stimulatory role of cancer cells on CB2 expression in macrophages was also validated in vitro experiments (Supplementary Fig. 4a).

According to aforementioned results, someone would presume that MGLL and CB2 might regulate each other's expression in TAMs. However, we found that CB2 expression was not affected by the MGLL transgene in TAMs (Supplementary Fig. 4b). We further constructed another transgenic mouse model with myeloid-specific overexpression of CB2 (Supplementary Fig. 4c, d). We also verified that CB2 did not regulate MGLL expression in MC-38 TAMs (Supplementary Fig. 4e). Therefore, the

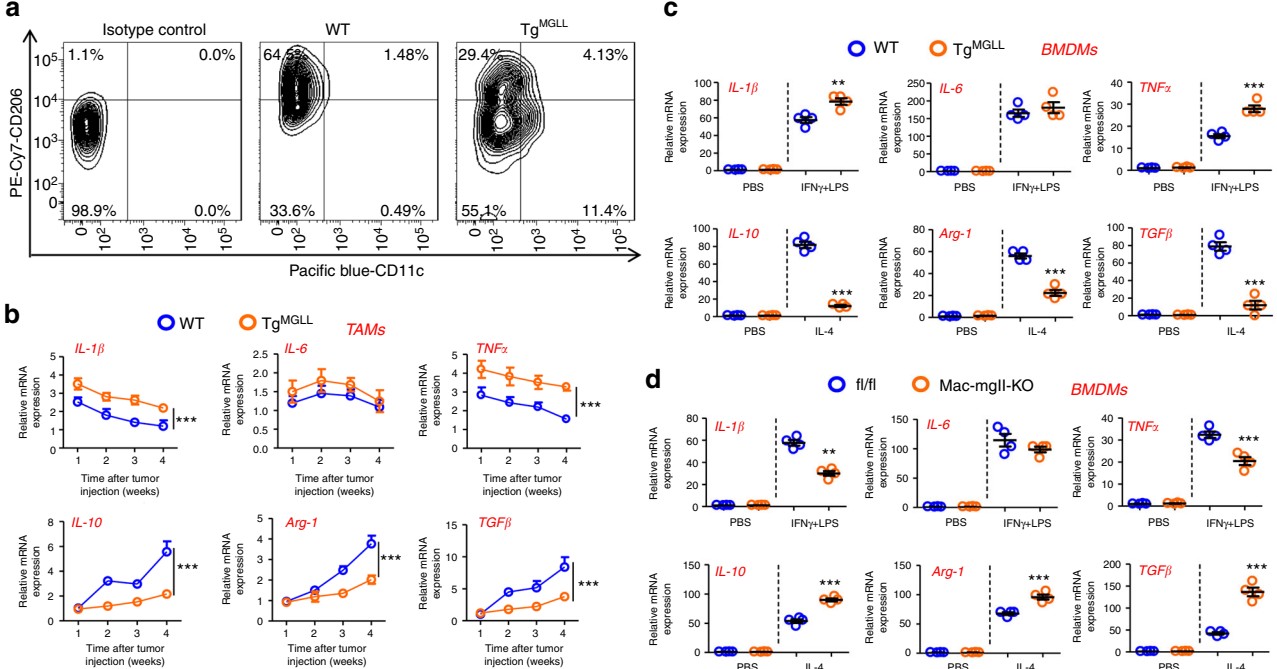

**Fig. 3** MGLL potentiates proinflammatory cytokine expression in tumor-associated macrophages. **a** Quantification of M1-like and M2-like TAMs in MC-38 tumors which were subcutaneously inoculated for 3 weeks in WT or Tg$^{MGLL}$ mice. Each tested sample was pooled from five individual ones. The M1-like TAMs were assed as CD3-CD45R-Gr1-GFP-CD45+F4/80+CD11c+CD206-; The M2-like TAMs were assed as CD3-CD45R-Gr1-GFP-CD45+F4/80 +CD206+CD11c-. The inoculated MC-38 cells were stably marked with GFP. This test was repeated three times. Representative results were displayed. **b** mRNA expression of cytokines, as indicated, in the TAMs from WT or Tg$^{MGLL}$ mice were measured dynamically ($n = 5$, ***$P < 0.005$). **c** MGLL potentiates M1 and blocks M2 activation of macrophages. The BMDMs from WT or Tg$^{MGLL}$ mice were treated with IFNγ (10 ng ml$^{-1}$) + LPS (100 ng ml$^{-1}$) or IL-4 (10 ng ml$^{-1}$) for 6 h and then subjected to real-time PCR assays ($n = 4$, **$P < 0.01$, ***$P < 0.005$). **d** mRNA levels of cytokines in the fl/fl or Mac-mgll-KO mice-derived BMDMs treated with IFNγ (10 ng ml$^{-1}$) + LPS (100 ng ml$^{-1}$) or IL-4 (10 ng ml$^{-1}$) for 6 h ($n = 4$, **$P < 0.01$, ***$P < 0.005$). Data in **b–d** represent the means ± s.e.ms. (Student's $t$-test)

expression of MGLL and CB2 in TAMs was independently regulated by the tumor microenvironment.

The increases in 2-AG levels and CB2 expression in TAMs indicated that endogenous cannabinoid signaling might be activated by MGLL deficiency to regulate macrophage polarization. As expected, Tg$^{MGLL}$-potentiated expression of IL-1β and TNFα in BMDMs with IFNγ and LPS stimulation was prevented by the addition of a CB2 transgene (Fig. 4d). Similarly, Tg$^{MGLL}$-suppressed expression of IL-10, Arg-1, and TGFβ in BMDMs stimulated with IL-4 was fully rescued by the additional CB2 transgene (Fig. 4e). In MC-38 tumor models, we demonstrated that Tg$^{MGLL}$ stimulated the expression of proinflammatory cytokines (IL-1β and TNFα) and reduced the levels of anti-inflammatory cytokines (IL-10, Arg-1, and TGFβ) in a CB2-dependent manner in TAMs (Supplementary Fig. 4f). However, the mechanism linking CB2 to cytokine production needs further to be elucidated.

TLR4 mediates the activation of inflammatory pathways like NF-κB and JNK signaling, which are important modulators of macrophage activation and cytokine production[8, 19]. The specific ligands of TLR4 include free fatty acid and lipopolysaccharide (LPS)[8]. Therefore, we presumed that MGLL-CB2 pathway might regulate TLR4 signaling. As expected, we demonstrated that inactivation of CB2 with a specific antagonist AM630 could potentiate LPS-stimulated phosphorylation of p65 and JNK, while the endogenous CB2 ligand 2-AG could block LPS-stimulated TLR-4 activation (Fig. 4f). Interestingly, we found that CB2 co-localized with TLR4 in macrophages in a basal condition in immunofluorescence analysis (Fig. 5a), indicating a protein–protein interaction, which was confirmed by

immunoprecipitation test (Fig. 5b). We further demonstrated that the protein–protein interaction between TLR-4 and CB2 could be attenuated by LPS or AM630 treatment and potentiated by 2-AG stimulation in primary macrophages (Fig. 5b, c). Those results indicated that CB2 might be an endogenous brake of TLR4 signal transduction by directly binding to TLR4. Most convincingly, the anti-inflammatory function of CB2 in TAMs was abrogated with TLR4 deletion (Fig. 5d).

**MGLL/CB2 regulates tumor progression in mice**. We next explored whether macrophage MGLL/CB2 axis would regulate the activity of tumor-associated CD8+ T cells. As expected, in subcutaneous tumor models, macrophage Tg$^{MGLL}$ largely potentiated mRNA levels of TNFα, IL-6, and IFNγ in tumor-associated CD8+ T cells, and this effect was prevented by additional macrophage Tg$^{CB2}$ (Fig. 6a). Consistently, the MGLL transgene-mediated tumor suppression (tumor volume and survival) was reversed by the addition of a CB2 transgene (Fig. 6b, c). In intravenous injection models, Tg$^{MGLL}$-suppressed incidence of liver metastasis was also rescued by additional Tg$^{CB2}$ (Fig. 6d). In the pathological observation of lung metastasis, we also demonstrated that the macrophage CB2 transgene largely rescued MGLL transgene-reduced tumor lesions (Fig. 6e, f).

To further confirm the role of macrophage MGLL/CB2 axis in tumor progression, TAMs from WT, Tg$^{MGLL}$, Tg$^{CB2}$, and Tg$^{MGLL+CB2}$ mice bearing MC-38 tumors were isolated and implanted subcutaneously with tumor cells in the WT mice. The Tg$^{MGLL}$ TAMs suppressed cancer cell growth in comparison to the WT TAMs and this effect was interrupted by additional

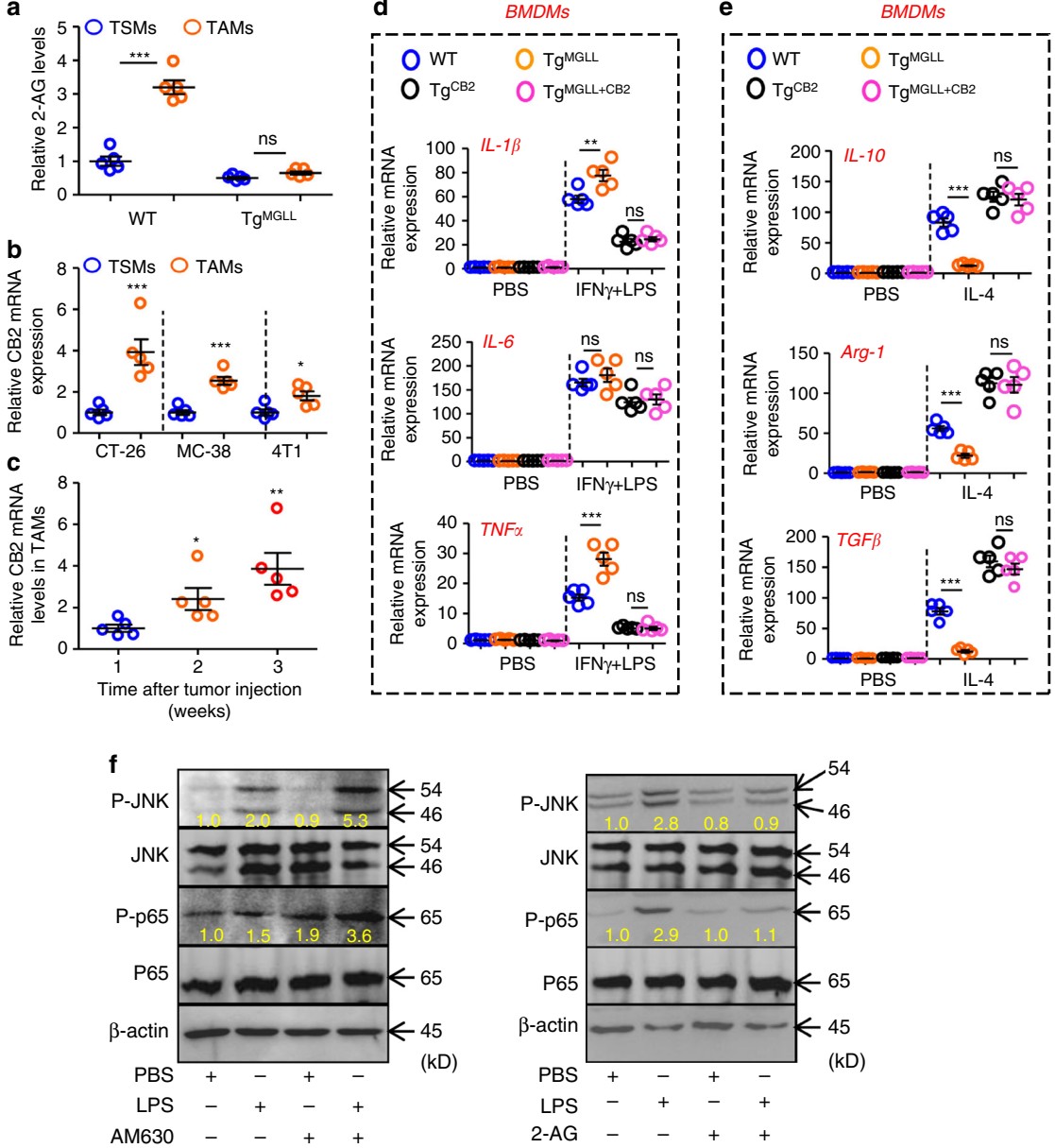

**Fig. 4** MGLL regulates macrophage activation via CB2 signal. **a** Relative 2-AG levels in TSMs and TAMs. Six-week-old mice were subcutaneously inoculated with MC-38 cells for 3 weeks. TSMs and TAMs were isolated for 2-AG assays. (*n* = 5) **b** Relative CB2 mRNA levels in TSMs and TAMs from 9-week-old mice inoculated with indicated subcutaneous tumors for 3 weeks. (*n* = 5) **c** Relative CB2 mRNA levels in TAMs from MC-38 tumor tissues as indicated time points after tumor injection in WT mice. (*n* = 5) **d** Relative mRNA expression of proinflammatory cytokines in the PBS- or IFNγ (10 ng ml$^{-1}$) + LPS (100 ng ml$^{-1}$)-treated BMDMs from the WT, Tg$^{MGLL}$, Tg$^{CB2}$ or Tg$^{MGLL+CB2}$ mice. (*n* = 5) **e** Relative mRNA levels of ant-inflammatory cytokines in the PBS- or IL-4 (10 ng ml$^{-1}$)-treated BMDMs from the mice described in **d**. (*n* = 5) **f** CB2 antagonizes TLR-4 signaling. Mouse peritoneal macrophages (PMs) were primed with PBS, a CB2 inhibitor AM630 (200 nM) or a CB2 activator 2-AG (10 μM) for 2 h, and then additionally treated with PBS or LPS (500 ng ml$^{-1}$) for 30 min before harvested for immunoblotting assays. The yellow values indicated the relative expression (P-JNK/JNK or P-p65/p65) according to the density. The expression value in control group for each interested protein was set as 1. Data in **a**–**e** are means ± s.e.ms. (*$P < 0.05$, **$P < 0.01$, ***$P < 0.005$; student's *t*-test; ns not significant)

transgene of CB2 in TAMs (Fig. 6g). However, whether MGLL/CB2 axis in other myeloid cells (dendritic cells, monocytes, etc.) would regulate tumor development is still not elucidated.

We also investigated the role of macrophage MGLL/CB2 axis in the mouse mammary tumor virus-driven polyoma virus middle T oncogene (MMTV-PyVT) mice, which spontaneously develop multiple primary tumors with lung metastases[20, 21]. We demonstrated that macrophage transgene of MGLL largely inhibited the primary tumor growth and lung metastases and this inhibitory effect could be diminished by

additional transgene of CB2 in macrophages (Supplementary Fig. 5a, b).

**CB2 antagonism suppresses tumor progression in mice**. The above results suggested the importance of macrophage CB2 in promoting cancer development. It is worth determining whether CB2 could be used as a therapeutic target because a series of CB2 agonists or antagonists have been designed[22–24]. Two specific antagonists of CB2 (AM-630 and JTE-907) were tested in the

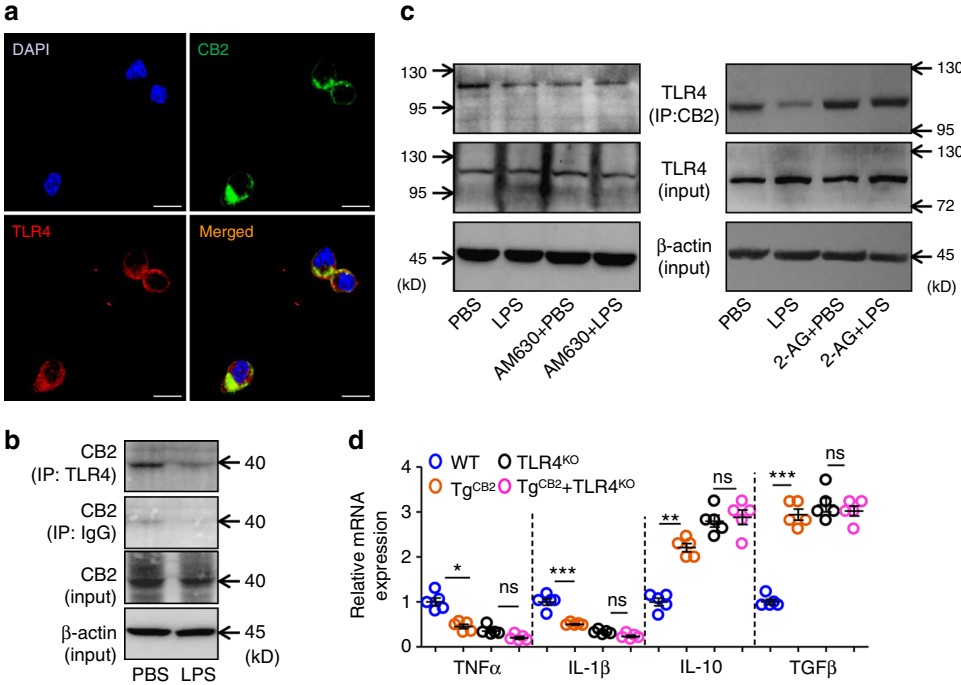

**Fig. 5** CB2 antagonizes TLR-4 signaling in macrophages. **a** The PMs were double-stained with TLR4 (Red) and CB2 (Green) antibodies by immunofluorescence. The nucleus was visualized by DAPI staining (Blue). This test was repeated three times. Representative images were shown. Scale bars, 40 μm. **b** Protein–protein Interaction between CB2 and TLR4 in the mouse peritoneal macrophages treated with PBS or LPS (500 ng ml$^{-1}$) for 30 min. The proteins were immunoprecipitated with TLR4 antibody or IgG as control. This test was repeated three times. Representative results were displayed. **c** LPS or AM630 inhibits, while 2-AG induces the Interaction between CB2 and TLR4 in macrophages. The mouse PMs described in **f** were harvested and the proteins were immunoprecipitated with CB2 antibody. **d** Relative mRNA levels of inflammatory cytokines in the TAMs from the WT, Tg$^{CB2}$, TLR4$^{KO}$ or Tg$^{CB2}$ + TLR4$^{KO}$ mice bearing MC-38 tumors for 3 weeks. ($n = 5$) Data in **j** are means ± s.e.ms. (*$P < 0.05$, **$P < 0.01$, ***$P < 0.005$; student's $t$-test; ns not significant)

subcutaneous tumor models. We demonstrated that treatment with AM-630 or JTE-907 could largely delay the growth of MC-38 tumors (Fig. 7a) and improve the survival of tumor-bearing mice (Fig. 7b). The anti-tumor effects (Improvement of tumor growth and mouse survival) of AM-630 and JTE-907 were confirmed in 4T-1 (Supplementary Fig. 6a, b) and CT-26 tumor models (Supplementary Fig. 6c, d). More convincingly, AM630 could also suppress the primary tumor growth and lung metastases in MMTV-PyVT mouse models (Fig. 7c, d).

To exclude the direct action of CB2 antagonists on cancer cells in drug treatment, CB2 in cancer cells was knocked out before inoculation (Supplementary Fig. 6e). We demonstrated that the CB2-deficient MC-38 tumors could also be inhibited by CB2 antagonists (AM-630 and JTE-907) in vivo (Supplementary Fig. 6f).

**MGLL in TAM predicts the survival of CRC patients**. To correlate the aforementioned findings to physiopathology in the clinic, we measured the expression of MGLL, CB2, and TGFβ in macrophages from the adjacent normal or carcinoma tissues. We performed immunohistochemical staining in specimens from patients with colorectal cancer (CRC) and found that MGLL expression in macrophages in carcinoma tissues was much lower than that in adjacent normal tissues (Fig. 8a, b). In addition, we isolated macrophages from the fresh carcinoma tissues and adjacent normal tissues of patients with CRC and verified that the mRNA expression of macrophage MGLL in carcinoma tissues was much lower than that in adjacent normal tissues (Fig. 8c). Meanwhile, the mRNA expression of macrophage CB2 and TGFβ was notably higher than that in the adjacent normal tissues (Fig. 8d, e). Interestingly, the higher levels of MGLL or lower

levels of CB2 in TAMs of patients with CRC predicted better survival (Fig. 8f, g).

## Discussion

The mechanism and function of lipid accumulation in TAMs are obscure. In the present study, we revealed that lipid accumulation in TAMs resulted from MGLL deficiency. MGLL deficiency led to macrophage activation toward an M2-like phenotype via endogenous 2-AG-CB2 cannabinoid signaling in TAMs. We also uncovered the direct interaction between CB2 and TLR4 and deciphered its role in macrophage activation. The TAM MGLL-CB2 axis regulated the activation of tumor-associated CD8+ T cells and the progression of multiple cancers. Targeting CB2 in macrophages delayed cancer progression (Supplementary Fig. 7).

Previous studies have revealed that tumor-associated dendritic cells are full of triglyceride-rich lipid droplets due to increased uptake of extracellular lipids by scavenger receptor A[12]. Here, we demonstrated a novel mechanism of lipid deposition in TAMs. Together with our previous study[2], we revealed a full picture of lipid reprogramming in TAMs: ABHD5 was increased, whereas MGLL was deficient. According to the biochemical activity of these two enzymes, MGLL deficiency might lead to an increase in monoglycerides[25], whereas ABHD5 upregulation might result in a decrease in triglycerides and an increase in diglycerides[9]. However, here, we demonstrated that the levels of triglycerides, diglycerides, and monoglycerides were all upregulated in TAMs. We presumed that the levels of cellular triglycerides might be elevated by the hydrophobic lipid droplets, which were primarily formed by diglycerides and monoglycerides. However, the lipid droplets could not be stored or enlarged unlimitedly. Therefore, ABHD5, a lipolytic factor, stood out as a factor to fuel the cells.

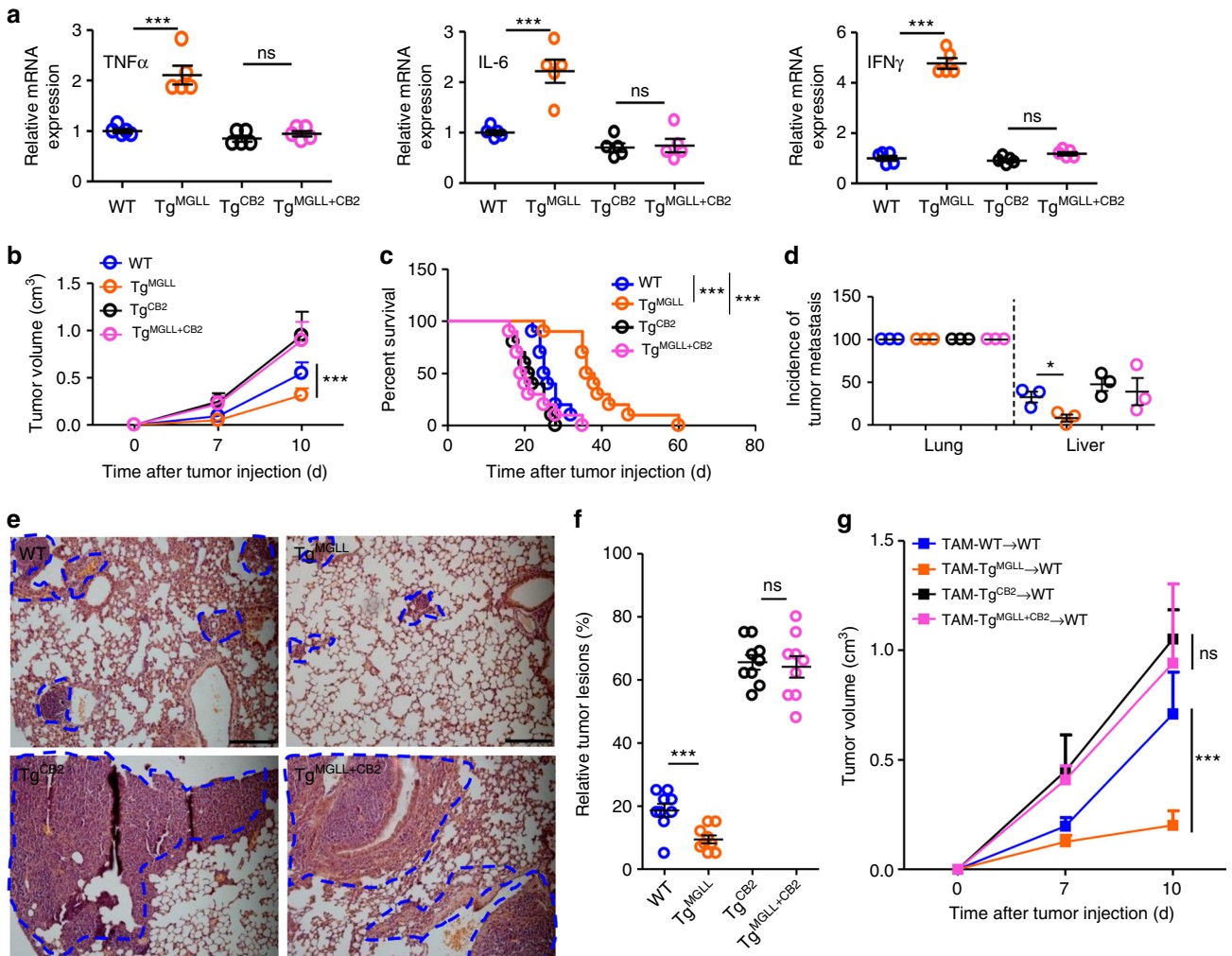

**Fig. 6** MGLL-CB2 axis regulates tumor progression in mice. **a** Relative mRNA levels of TNFα, IL-6 and IFNγ in the CD8+ T cells in the MC-38 tumors inoculated subcutaneously in WT, Tg$^{MGLL}$, Tg$^{CB2}$, and Tg$^{MGLL+CB2}$ mice. The data represent the means ± s.e.ms. ($n = 5$, ***$P < 0.005$, Student's $t$-test; ns not significant). **b** Six-week-old WT, Tg$^{MGLL}$, Tg$^{CB2}$, and Tg$^{MGLL+CB2}$ mice were subcutaneously inoculated with MC-38 cells, and the tumor volume was measured dynamically. This experiment was repeated three times. The data represent the means ± s.e.ms. ($n = 10$, **$P < 0.01$, ***$P < 0.005$, Student's $t$-test). **c** The survival time of the MC-38 tumor-bearing mice. The mice were described in **b**. The data represent the means ± s.e.ms. ($n = 10$, **$P < 0.01$, ***$P < 0.005$, Gehan-Breslow-Wilcoxon test). **d** The incidence of tumor metastasis in MC-38 tumor model. Six-week-old WT, Tg$^{MGLL}$, Tg$^{CB2}$, and Tg$^{MGLL+CB2}$ mice were intravenously injected with MC-38 cells via the tail vein, and the metastases in the lungs and livers were checked 3 weeks later. This experiment was repeated three times. The data represent the means ± s.e.ms. ($n = 6$–$10$, *$P < 0.05$, Student's $t$-test). **e** H&E staining of the lungs from the MC-38 tumor-bearing WT, Tg$^{MGLL}$, Tg$^{CB2}$, and Tg$^{MGLL+CB2}$ mice as described in **d**. The blue dotted lines indicate the tumor lesions. Scale bars, 200 μm. **f** Relative tumor lesions were calculated. The data represent the means ± s.e.ms. ($n = 9$, ***$P < 0.005$, Student's $t$-test). **g** Volume of tumors from WT mice implanted with MC-38 cells admixed 2:1 with macrophages isolated from corresponding tumors grown in WT, Tg$^{MGLL}$, Tg$^{CB2}$ or Tg$^{MGLL+CB2}$ mice. The data represent the means ± s.e.ms ($n = 5$, ***$P < 0.005$, Student's $t$-test; ns not significant)

We concluded that ABHD5 upregulation might be necessary for the survival of TAMs, whereas MGLL was the major contributor to lipid accumulation in TAMs. It should be noted that how the lipid metabolic gene MGLL is regulated by cancer cells is still obscure.

A previous study demonstrated that MGLL was elevated in carcinoma tissues and promoted cancer pathogenesis via regulation of the fatty acid network[26]. However, we found that MGLL expression in TAMs was reduced and that TAM MGLL functioned as a tumor suppressor. It is interesting that the expression of MGLL in TAMs and cancer cells was contradictory, although those two types of cells were in the same microenvironment. A similar phenomenon was also observed in our previous study regarding the expression of ABHD5 in cancer cells and TAMs[2]. We speculate that cancer cells and the associated stromal cells in

the same microenvironment might exhibit opposite expression of certain metabolism-related genes. The tumor microenvironment is innutritious, and cancer cells are generally dominant in the nutrient competition. Thus, stromal cells lack some important nutrients (for example, glucose, fatty acids, and amino acids), and metabolic pathways might be reprogrammed to support cell survival. However, the precise regulatory mechanisms of gene expression in the tumor microenvironment require further exploration in future studies.

It is also interesting that MGLL in cancer cells promoted tumor progression by releasing special fatty acids[26], whereas MGLL in TAMs suppressed cancer development by attenuating endogenous CB2 cannabinoid signaling. Therefore, it is not wise to interrupt cancer progression by simply modulating the expression or activity of MGLL because MGLL-related drugs are not specific

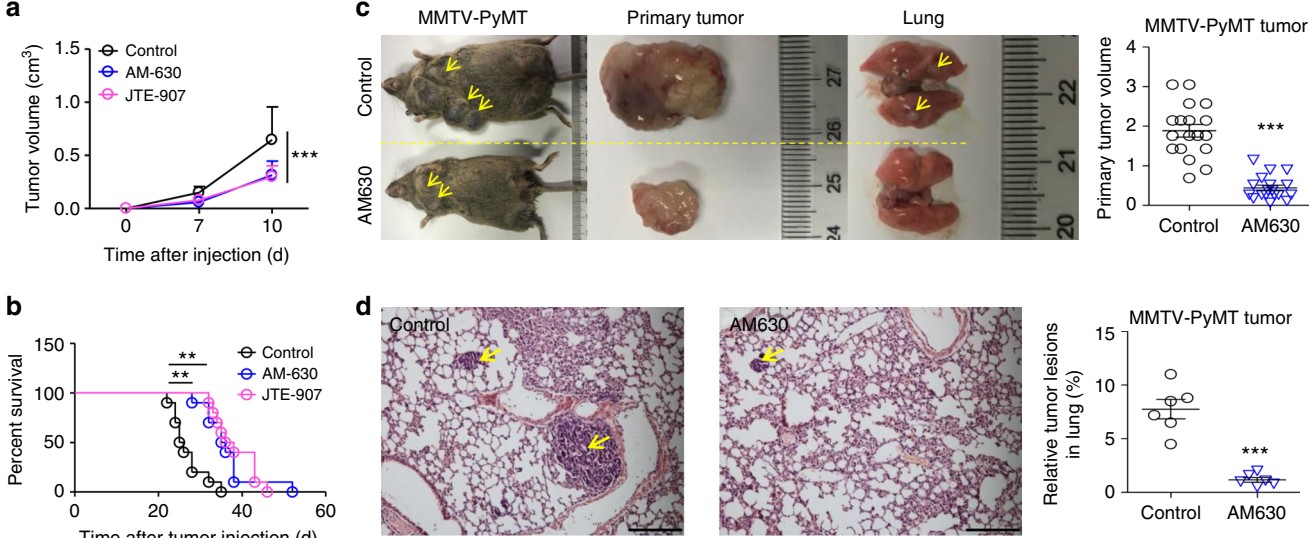

**Fig. 7** Targeting CB2 delays tumor progression. **a** CB2 antagonists inhibit tumor growth. Six-week-old mice were subcutaneously inoculated with MC-38 tumors and treated with CB2 antagonists, AM-630 (0.3 mg kg$^{-1}$ d$^{-1}$, i.p.) or JTE-907 (0.5 mg kg$^{-1}$ d$^{-1}$, i.p.). The tumor volume was measured dynamically. The data represent means ± s.e.ms. ($n = 10$, ***$P < 0.005$, Student's $t$-test). **b** CB2 antagonists improve the survival of tumor-bearing mice. The MC-38 tumor-bearing mice were described in **a**. The survival time was recorded. The experiment was repeated twice ($n = 10$, **$P < 0.01$, Gehan-Breslow-Wilcoxon test). **c** The MMTV-PyMT mice were treated with PBS or AM630 (0.3 mg kg$^{-1}$ d$^{-1}$, i.p.) starting from 6 weeks old and sacrificed at 14 weeks old. The representative gross morphology of primary mammary tumors and lung metastasis were displayed. The size of the primary mammary tumors were calculated. ($n = 6$, ***$P < 0.005$, Student's $t$-test). **d** Tumor lesions in lungs from the mice described in **c** ($n = 6$, ***$P < 0.005$, Student's $t$-test). Scale bars, 200 μm

to macrophages or cancer cells. Future studies are needed to test the effects of systemic administration of MGLL-related drugs on specific types of cancers. Luckily, downstream of MGLL in macrophages, we identified a precise and effective target, CB2, which is also an oncogenic factors in tumor cells[27, 28]. Therefore, as shown in the present study, targeting CB2 notably suppressed tumor growth and improved survival by sustaining the activity of tumor-associated CD8+ T cells.

It should be emphasized here that CB2 worked as a brake of TLR4, and MGLL functioned as a two-way switch in the regulation of macrophage activation via CB2/TLR4 interaction. When macrophage MGLL was upregulated or activated, decrease of 2-AG would discharge the CB2-suppressed TLR4 signal, and simultaneously increased free fatty acid might activate TLR4 signal transduction. When macrophage MGLL was deficient (For example, in tumor microenvironment), increase of 2-AG would activate the CB2-mediated TLR4 suppression, and simultaneously free fatty acid-stimulated TLR4 would also be restricted (Supplementary Fig. 7).

Collectively, we demonstrated that MGLL functioned as a CB2-dependent switch in regulating the activation of TAMs, exhaustion of CD8+ T cells and subsequent development of cancers. Our findings shed light on therapeutic strategies to treat those malignances.

## Methods

**Cell culture**. MC-38 cells were provided by JENNIO Biological Technology (Guangzhou, China) and maintained in our research group. The Raw264.7 macrophage-like cell line (RAW cells) and CT-26 and 4T1 cells were purchased from ATCC (Rockville, MD, USA). All cells had been authenticated and tested for mycoplasma. All the cell lines and primary mouse macrophages were grown in DMEM or 1640 medium supplemented with 10% fetal bovine serum (FBS) at 37 °C in a humidified 5% CO$_2$ atmosphere.

**Establishment of stable cell lines**. pCDNA-MGLL RAW cells were established with stable overexpression of murine MGLL, and the control cells were pCDNA3.1-transfected macrophages. The MGLL cDNA was cloned from an Origene cDNA clone (MC202347). The gene *cnr2* (Protein: CB2) in the MC-38

mouse cancer cells was knocked out using a CB2 CRISPR/Cas9 KO plasmid (sc-419723). The establishment of stable cell lines was verified by immunoblotting assays of target proteins.

**Genetically-engineered mouse models**. These mouse studies were approved by the Institutional Animal Care and Use Committee of Third Military Medical University and were performed in accordance with relevant guidelines. All mice were housed in a specific pathogen-free environment.

The FVB/N-Tg(MMTV-PyVT)/Nju mice (#N000228) which spontaneously develop primary mammary tumors with lung metastases[20, 21], the C57BL/10ScNJNju mice (#000192) with the knockout of *tlr4* gene[29, 30], and the C57BL/6JNju Rag-1-deficient mice (#N000013) which have no mature lymphocytes[31], were provided by National Model Animal Resource Information Platform (Nanjing University, China).

The myeloid specific transgene or knockout mouse models mediated by CD11b or lysozyme promoter are commonly used to investigate the in vivo function of a specific gene in macrophages[2, 10, 32, 33]. The cDNA of mouse *mgll* was subcloned into a transgenic construct containing the human CD11b promoter to drive myeloid cell-specific gene expression. The myeloid specific *mgll* transgenic construct was microinjected into C57BL/6 embryos according to standard protocols, and the founders were crossed with the wild-type C57BL/6 strain. The line with the highest level of mgll expression in macrophages (Tg$^{MGLL}$) was selected for further study.

Using the same protocol described above, the cDNA of mouse *cnr2* (CB2) was subcloned, and myeloid specific *cnr2* transgenic (Tg$^{CB2}$) mice were obtained. Tg$^{MGLL}$ mice were mated with Tg$^{CB2}$ mice to obtain the double transgenic (Tg$^{MGLL+CB2}$) mice.

The *mgll* gene (NM_001166251.1) contains 8 exons. Exon 3 was selected as a conditional knockout (cKO) region. To engineer the targeting vector, homology arms and the cKO region were generated by PCR using BAC clones RP23–277O5 and RP23–220I16 from the C57BL/6J library as templates. In the targeting vector, the Neo cassette was flanked by Rox sites, and the cKO region was flanked by LoxP sites. Diphtheria toxin was used for negative selection. The constitutive KO allele was obtained after Cre-mediated recombination. C57BL/6 ES cells were used for gene targeting. Myeloid cell-specific mgll knockout (Myeloid-mgll-KO) mice were generated by crossing MGLL-floxed mice to mice expressing the lysozyme promoter-driven Cre recombinase (#004781, Jackson Laboratory Stock).

**Subcutaneous tumor models**. This model was set up as described in our previous work[2, 34]. Briefly, six-week-old WT or genetically-engineered C57BL/6 or BALB/c mice were maintained in our lab. The BALB/c mice were subcutaneously injected with CT-26 colorectal cancer cells or 4T1 breast cancer cells (5 × 10$^6$ cells per mouse), and the wild-type or genetically-engineered C57BL/6 mice were inoculated

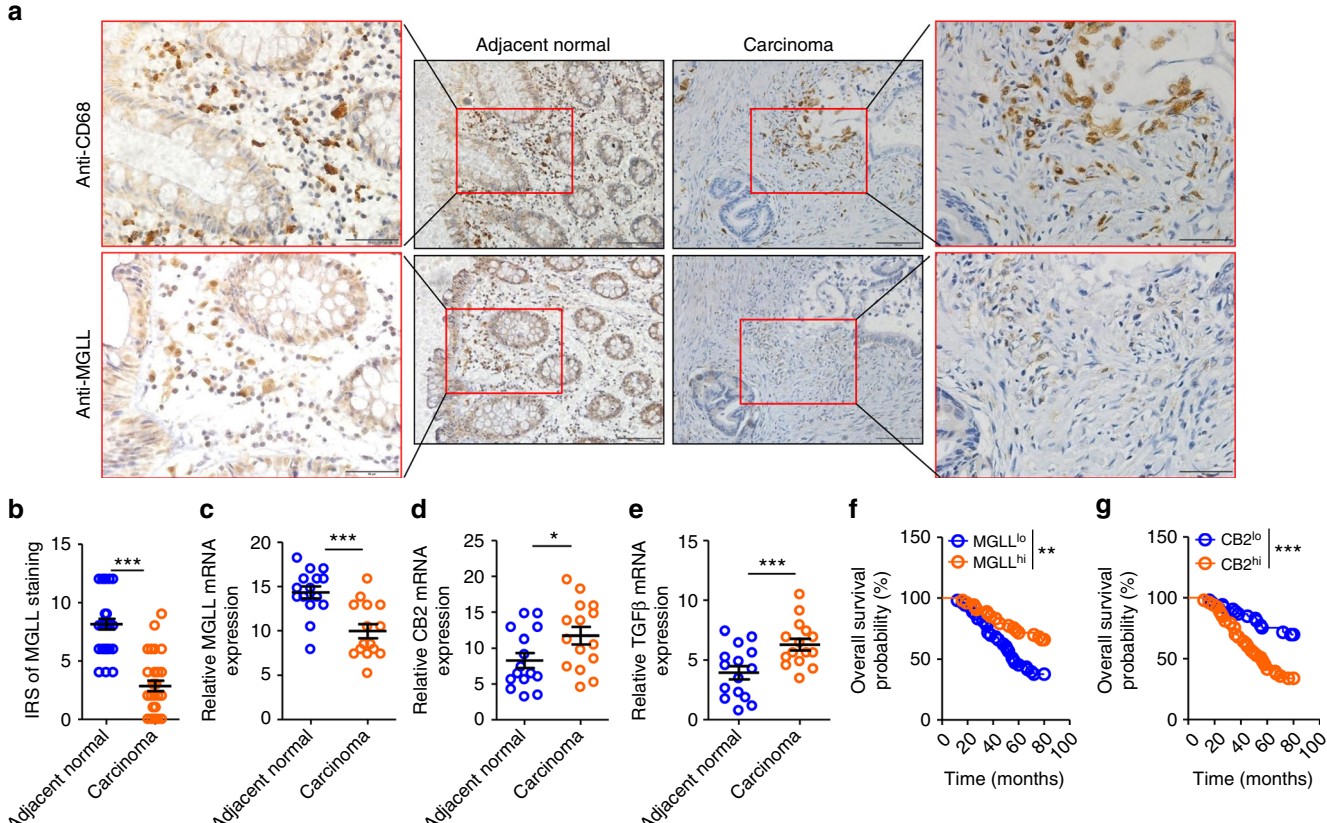

**Fig. 8** Expression of macrophage MGLL is decreased in carcinoma tissues and positively correlated to the survival of patients with CRC. **a** Representative images from immunohistochemical staining of CD68+ cells (macrophages) in adjacent normal tissue and carcinoma tissue. The corresponding area of infiltrated macrophages was stained with MGLL antibody on the consecutive slides. MGLL expression in macrophages in carcinoma was much lower than that in adjacent normal tissues. The representative areas of macrophages at higher magnification are displayed in the red square areas. The scale bar at higher magnification represents 50 μm and at lower magnification represents 200 μm. **b** The statistical data were obtained from immunohistochemical staining analysis of 30 samples from 30 patients. The data represent the means ± s.e.ms. ($n = 30$, ***$P < 0.005$, Student's $t$-test). **c–e** The relative mRNA expression of MGLL (**c**), CB2 (**d**), and TGFβ (**e**) in macrophages from the carcinoma tissues or adjacent normal tissues. The macrophages were isolated from fresh carcinoma tissues or adjacent normal tissues from CRC patients who underwent surgery, and the macrophages were subjected to real-time PCR assays. The data represent the means ± s.e.ms. ($n = 15$, *$P < 0.05$, ***$P < 0.005$, Student's $t$-test). **f–g** Overall survival of patients with CRC with differential expression of MGLL (**f**) or CB2 (**g**) in TAMs. The investigated patients were classified into two groups according to the mRNA expression of MGLL or CB2 in TAMs. The 50% of patients with high TAM MGLL or CB2 expression were assigned to group MGLL[hi] or CB2[hi], respectively. The remaining patients with low TAM MGLL or CB2 expression were assigned to group MGLL[lo] or CB2[lo], respectively. The overall survival time of the patients were obtained via follow-up visit ($n = 53$, **$P < 0.01$, ***$P < 0.005$, Gehan-Breslow-Wilcoxon test)

with MC-38 colorectal cancer cells ($5 \times 10^6$ cells per mouse) in their thighs. The tumor volume (size) was measured dynamically and calculated as 0.523 × (length × width × height). This study was approved by the Institutional Animal Care and Use Committee of the Third Military Medical University and was performed in accordance with relevant guidelines. Subcutaneous models would be more valuable for breast cancer study because the cancer cells could be injected orthotopically and conveniently.

**Intravenous tumor models.** The 6-week-old C57BL/6 WT or transgenic mice were intravenously injected with MC-38 colorectal cancer cells ($5 \times 10^6$ cells per mouse) via the tail vein. The mice were sacrificed 2 weeks after tumor inoculation. The lungs and livers were dissected for pathological observation of tumor lesions with H&E staining. This study was approved and performed in accordance with relevant guidelines of the Third Military Medical University.

**Treatment of antibodies in vivo.** The MC-38 tumor-bearing mice were treated with monoclonal antibodies anti-CD8 (MAB116, Novus Biologicals) or IgG (MAB002, Novus Biologicals) as control. Tumor cells were inoculated on day 1. In the experiment investigating the survival time of the tumor-bearing mice, each mouse was treated three times (days 4, 7, and 10) via the tail vein at a dose of 100 μg/day.

**Cell viability assay.** A CCK8 assay was performed according to the manufacturer's protocol (C0037, Beyotime, China). MC-38 cells (2000 cells per 100 μl of medium) were plated in 96-well plates. At different time points, 10 μl of CCK8 solution was added to each well, and the wells were cultured at 37 °C in a humidified 5% $CO_2$ atmosphere for 1 h. Then, the absorbance at 450 nm of each well was measured.

**Real-time PCR.** Total RNAs were extracted using a kit provided by Thermo Fisher Scientific (#10296010). RNAs were transcribed into cDNAs using PrimeScript (DRR047A, Takara, Dalian, China). qPCR was performed using a 7900HT Fast Real-Time PCR system or ABI 7500 Real-Time PCR system (Applied Biosystems, Darmstadt, Germany). Expression levels were normalized to β-actin. Reactions were performed in duplicate using Tli RNaseH plus and universal PCR master mix (#RR820A, TakaRa). The relative expression was calculated by the $2^{(-DDCt)}$ method. The primers were displayed in Supplementary Table 1.

**Western blotting.** Tissue and cell proteins were extracted using RIPA Lysis Buffer (#P0013, Beyotime, China) and quantified using a BCA kit (#P0009, Beyotime, China). Fifty micrograms of each protein sample was separated by 8% or 10% SDS-PAGE and transferred to a polyvinylidene difluoride membrane. The membranes were blocked with 5% BSA and incubated with primary antibodies for 10 h at 4 °C. The membranes were rinsed five times with PBS containing 0.1% Tween 20 and incubated for 1 h with the appropriate horseradish peroxidase-conjugated secondary antibody at 37 °C. Membranes were extensively washed with PBS containing 0.1% Tween 20 three times. The signals were stimulated with enhanced chemiluminescence substrate (#NEL105001 EA, PerkinElmer) for 1 min and detected with a Bio-Rad ChemiDoc MP System (170–8280). The primary

antibodies included anti-MGLL (#Sab-2500641, Sigma; the dilution ratio was 1:1000), anti-CB2 (#ab3561, Abcam; the dilution ratio was 1:1000), anti-TLR4 (#sc-293072, Santa Cruz, the dilution ratio was 1:1000), anti-Phospho-NF-κB p65 (Ser536) (P-p65) (#3033S, Cell Signaling, the dilution ratio was 1:2000), anti-Phospho-SAPK/JNK (Thr183/Tyr185) (P-JNK) (#4668S, Cell Signaling, the dilution ratio was 1:1000), anti-SAPK/JNK (#9252, Cell Signaling, the dilution ratio was 1:1000), anti-p65 (#GTX107678, GeneTex, the dilution ratio was 1:1000), anti-GAPDH (#2118, Cell Signaling; the dilution ratio was 1:2000), anti-β-actin (#3700, Cell Signaling; the dilution ratio was 1:2000), and anti-Tubulin-α (#3873P, Cell Signaling, the dilution ratio was 1:2000). The primary Images (Supplementary Figs. 8–9) were cropped for presentation.

**Immunohistochemistry of patient samples.** Thirty cases of formalin-fixed and paraffin-embedded CRC samples in this study were obtained from the tissue bank of the Department of Oncology at Southwest Hospital at the Third Military Medical University. Those patients included 16 males and 14 females. The average age was 58.2 years. The TNM state was $T_{2-3}N_0M_0$. All the colorectal cancer tissues were primary and untreated before surgery, and the specimens were collected from 2007 to 2010. Informed consent was obtained from all the subjects. Tumor tissues were collected in compliance with the regulations approved by the Scientific Investigation Board of the hospital. All tissue slides were de-waxed and rehydrated. The slides were then incubated in 0.3% $H_2O_2$ in methanol for 30 min to block endogenous peroxidase activity. Antigens were retrieved with 10 mmol $L^{-1}$ sodium citrate (pH 6) for 5 min in a pressure cooker. The slides were then incubated with the selected antibodies (anti-CD68, NB100–683, Novus Biologicals, the dilution ratio was 1:1000; anti-MGLL, #ab152002, Abcam, the dilution ratio was 1:1000) at 4 °C overnight. The slides without treatment with primary antibody served as negative controls. The slides were developed with an EnVisionTM method (DAKO, Capinteria, CA), visualized using diaminobenzidine solution, and then lightly counterstained with hematoxylin. Evaluation of immunohistochemical staining reaction was performed in accordance with the immunoreactive score (IRS) proposed by Remmele and Stegner[35]: IRS = SI (staining intensity) × PP (percentage of positive cells). Negative SI, 0; Weak SI, 1; Moderate SI, 2; Strong SI, 3. Negative PP, 0; 10% PP, 1; 11–50% PP, 2; 51–80% PP, 3; and >80% PP, 4. Ten microscopic fields (100×) from different areas of each tissue section were used for the IRS evaluation. Slides were examined and scored independently by three pathologists who were blinded to the patient information.

**Immunofluorescence staining.** The macrophages on the coverslips were fixed in 4% ice-cold paraformaldehyde in PBS for 20 min, washed with PBS three times (5 min each), and incubated for 30 min at room temperature in a protein-blocking solution. The coverslips were incubated with the primary antibodies (anti-CB2, #ab3561, Abcam, the dilution ratio was 1:1,000; anti-TLR4, #sc-293072, Santa Cruz, the dilution ratio was 1:1000) for 1 h at 37 °C and then at 4 °C overnight. After being washed, the coverslips were incubated at 37 °C for 1 h with fluorescent second antibodies (#111-545-003 or #715-606-150, Jackson ImmunoResearch). The cells were then counterstained with 40,6-diamidino-2-phenylindole to reveal cell nuclei. The specificity of the primary antibody was verified by omitting that antibody in the reaction.

**Staining of cellular lipids with Bodipy.** The cultured cells (Raw264.7 cells or macrophages) were fixed in 4% paraformaldehyde in PBS for 10 min at room temperature. After washed in PBS twice (2 min each), the fixed cells were stained with Bodipy493/503 at 0.5 μg $ml^{-1}$ in PBS for 15 min at 20 °C. The cells were counterstained with 4′,6-diamidino-2-phenylindole (DAPI) to reveal cell nuclei. Then, the cells were washed twice with PBS and observed under an inverted fluorescent microscope.

**Extract and measurement of cellular lipids.** Raw264.7 cells or extracted primary macrophages (spleen macrophages and tumor-associated macrophages) were placed into 3 ml of a mixture of chloroform and methanol [CHCl₃:MeOH (2:1)] overnight. Then, the tube was centrifuged at 2700 rpm for 15 min at room temperature, and the lipid extract was transferred into a clean 16 × 100 mm glass tube. The lipid extract was dried down under a $N_2$ stream on a heater at 60 °C and resuspended in 3 ml of the chloroform:methanol mixture before phase separation by addition of 0.6 ml of diluted $H_2SO_4$ (0.05%). The sample tube was centrifuged at 2000 rpm for 15 min at room temperature. The top aqueous phase and the middle protein phase were aspirated. The total volume of the bottom phase was recorded and transferred to a new 16 × 100 mm screw cap tube containing 1 ml of 1% Triton-X-100 in CHCl₃. This mixture was dried and then dissolved in 0.5 ml of water for lipid measurements by high-performance liquid chromatography (Biotree, Shanghai, China).

**Preparation of conditioned medium.** The mouse PMs were cultured in 250 ml flasks in regular medium (DMEM supplemented with 10% FBS). At the time of 80% confluence, 10 ml of DMEM with 1% FBS was added to each flask and was re-collected 48 h later to obtain macrophage-primed medium (MPM). The conditioned medium was obtained by mixing the MPM with the regular medium (v/v = 1:1). The conditioned medium was used to treat the cultured MC-38 cells in

transwell assays and cell scratch tests. It should be pointed out that the serum in MPM was almost depleted and thus the conditioned medium contained around 5% FBS.

**Transwell assays.** The migration ability of cancer cells was assessed using Transwell chambers with polycarbonate membrane filters and 24-well inserts (6.5 mm diameter and 8 μm pore size) (Corning Life Sciences, Corning, NY, USA). A total of 5000 cells in 150 μl of DMEM with antibiotics but without serum were seeded onto the upper chamber. The lower chamber was filled with 600 μl of conditioned medium of WT or Tg$^{MGLL}$ PMs. It should be noted that the serum free medium in upper chamber could eliminate the effects of cell proliferation on migration tests. The low serum could also slow down the migration rates and make the test more sensitive. The cancer cells in upper chamber could also survive, because the conditioned medium in lower chamber contained enough serum. The medium in both chambers was changed once daily. After the cells were cultured for 48 h, the non-migratory cells were removed from the upper surface of the filter using a Q-tip, and the migrated cells were fixed with chilled acetone (4 °C), stained with crystal violet solution (R40052, Thermo Fisher Scientific, Shanghai, China) and counted under ten different low-power (100×) microscopic fields. The cell border was verified by switching to the high-power objective lens (400×) during counting.

**Cell scratch tests.** MC-38 cells were cultured in 24-well plates. When the density reached 100%, a scratch was made across the monolayer of cells. After being washed with PBS three times, the cells were treated with conditioned medium from the WT or Tg$^{MGLL}$ PMs. Images of the scratches were captured on day 1 and day 3 using a digital camera (C5060, Olympus, Tokyo, Japan) mounted on an inverted microscope (CKX41, Olympus). Four different fields from each sample were considered for quantitative estimation of the distance between the borderlines, and in each image, four different equidistant points were measured to better estimate the true width of the wounded area. The migration rate is expressed as percentage (day 3 versus day 1), and it was calculated as the proportion of the mean distance between both borderlines caused by scratching to the distance that remained cell-free after re-growth of cells. The analysis was performed using software Image J.

**Isolation of peritoneal or bone marrow-derived macrophages.** The isolation protocol was described previously[2, 10]. Briefly, each mouse was injected intraperitoneally with 2 ml of 3% thioglycollate (Difco) on day 1 and sacrificed with $CO_2$ or isoflurane on day 3. After sacrifice, the mouse was injected intraperitoneally with 5 ml of DMEM containing 10% fetal bovine serum (FBS), penicillin and streptomycin, and the peritoneal cells were then collected into cell culture dishes. Two hours after culture, the floating cells were removed by washing the cells with PBS. The attached cells were considered PMs and subjected to the experiments.

The bone marrow medium (BMM) for BMDMs was made by mixing 30 ml of conditioned medium of the L929 murine fibroblast cell line with 70 ml of DMEM, containing 10% FBS and penicillin/streptomycin. On day 1, the mouse was sacrificed by $CO_2$, and the femur and tibia were separated. Each end of the bones was removed, and the bone marrow was washed out with a 25-gaugeneedle attached to a 10-ml syringe. The bone marrow-derived cells were centrifuged (500×$g$), resuspended in BMM and plated onto 6-well cell culture plates containing BMM at 4 ml per well. Fresh BMM (1 ml per well each time) was added to the cultured cells on day 3 and day 5. The medium was changed to DMEM containing 10% FBS and penicillin/streptomycin on day 6, and bone marrow-derived macrophages (BMDMs) were attached to the bottom of the culture dishes.

**Isolation of macrophages from spleens or tumor tissues.** Spleens from normal or tumor-bearing mice were dissected and put in fresh Buffer A containing PBS, 2 mM EDTA, and 0.5% BSA in a 60-mm dish. The spleen was gently rubbed between the two rough sides of frosted slides (#FSL006, Beyotime, Shanghai, China). The dissociated cells were collected into a 15-ml tube and centrifuged at 400 × $g$ for 5 min. The pellets were resuspended in 5 ml of ACK Lysing Buffer (#C3702, Beyotime, Shanghai, China), kept still for 5 min at room temperature, diluted to 15 ml with 10 ml of DMEM and centrifuged at 1000 × $g$ for 5 min. The ACK Lysis Buffer was then aspirated, and 1 ml of Buffer A was added to resuspend the pellets. The resuspended cells were filtered through a 100-μm filter (#08-771-19, Fisher) and spun at 400 × $g$ for 5 min. These cells were washed with Buffer A again and resuspended in an appropriate amount of Buffer A for further sorting of macrophages.

The fresh tumor tissues were cut into pieces and digested in Buffer A containing 1 g $L^{-1}$ type 4 collagenase (#LS004188, Worthington), 0.1 g $L^{-1}$ hyaluronidase (#H1115000, Sigma) and 0.01 g $L^{-1}$ DNase I (#D8071, Solarbio, China). The dissociated cells were collected into a 15-ml tube and centrifuged at 400 × $g$ for 5 min. The pellets were resuspended with ACK Lysing Buffer and washed with Buffer A before filtration with a 100-μm filter. These cells were collected for further isolation of macrophages and other immune cells.

**Immunostaining and fluorescence-activated cell sorting (FACS).** The splenocytes and tumor tissue-derived cells in Buffer A were counted. An aliquot of cells was pelleted and resuspended in the Sorting Buffer (PBS with 0.5% endotoxin-free

FBS, 2 mM EDTA, and 25 mM HEPES) at $10^7$ cells ml$^{-1}$. Cells were incubated with Fc Block (#553142, BD Biosciences) prior to staining with conjugated antibodies for 15 min at 4 °C, followed by two washes in Sorting Buffer. Cells were then resuspended in Sorting Buffer for FACS (FACSAria II or Accuri C6, BD Biosciences). The antibodies included PerCP/Cy5.5 anti-mouse CD45 antibody (#103132, Biolegend), anti-mouse F4/80 antigen APC (#123116, Biolegend), PE anti-mouse/human CD11b antibody (#101208, Biolegend), Pacific Blue™ anti-mouse CD11c (#117322, Biolegend), PE/Cy7 anti-mouse CD206 (MMR) antibody (#141720, Biolegend), FITC anti-mouse CD3 antibody (#100305, Biolegend), FITC anti-mouse/human CD45R/B220 antibody (#103205, Biolegend), FITC anti-mouse Ly-6G/Ly-6C (Gr-1) antibody (#108406, Biolegend), PE anti-mouse Ly-6G/Ly-6C (Gr-1) antibody (#108408, Biolegend), PE anti-mouse CD3 antibody (#100205, Biolegend), PE anti-mouse/human CD45R/B220 antibody (#103207, Biolegend), APC/Cy7 anti-mouse Ly-6G/Ly-6C (Gr-1) (#108424, Biolegend), PerCP/Cy5.5 anti-human CD45 antibody (#368504, Biolegend), APC anti-human CD68 antibody (#333809, Biolegend), and Pacific Blue™ armenian hamster IgG isotype control antibody. The markers for macrophage sorting were described previously[36, 37]. The tissue macrophages were assessed as CD3-B220-Gr1-CD45 +F4/80+. The M1-like macrophages were assessed as CD3-B220-Gr1-CD45+F4/80+ CD11c+CD206−. The M2-like macrophages were assessed as CD3-B220-Gr1-CD45+F4/80+ CD11c-CD206+. When the TAMs were isolated from the GFP-MC-38 tumors, the GFP+ cells were excluded. The macrophages in the human tumor tissues and adjacent normal tissues were assessed as CD45+CD68+.

**Isolation of macrophages from human tumor tissues**. The human tumor-associated macrophages were isolated and enriched from the patients in PLA 324 Hospital and Southwest Hospital in Chongqing (China). All the participants gave informed consent. All tumors were primary and untreated before surgery, and the specimens were anonymized. Tumor tissues were collected in compliance with the regulations approved by the Scientific Investigation Board of the hospitals. Hundred and six cases of colorectal cancer tissues and the corresponding adjacent normal tissues were collected. Those patients included 44 males and 62 females. The average age was 62.5 years. The TNM state was $T_{2-4}N_{0-2}M_{0-1}$ (Supplementary Table 2). The fresh tumor tissues were treated according to the method as described above in "Isolation of macrophages from spleens or tumor tissues". Finally, the macrophages (CD45+CD68+ cells) were isolated by flowcytometry (Accuri C6, BD Biosciences). The isolated macrophages were then subjected to RNA extraction and reverse transcription. The cDNAs were stored in at 80 °C before qPCR assays. All the participants were followed up. The investigator performing PCR assays was blinded to the patient information.

**Enrichment of CD4+ or CD8+ T cells from mouse tumor tissues**. The fresh tumor tissues were cut into pieces and digested in Buffer A containing 1 g L$^{-1}$ type 4 collagenase (#LS004188, Worthington), 0.1 g L$^{-1}$ hyaluronidase (#H1115000, Sigma) and 0.01 g L$^{-1}$ DNase I (#D8071, Solarbio, China). The dissociated cells were collected into a 15-ml tube and centrifuged at $400 \times g$ for 5 min. The pellets were resuspended with ACK Lysing Buffer and washed with Buffer A before filtration with a 100-µm filter. These cells were collected for further isolation of T cells. CD4+ T cells and CD8+ T cells were isolated using kits from eBioscience #8804-6821-74 and #8804-6822-74, respectively according to the provided protocols. The enriched CD4+ and CD8+ T cells were subsequently subjected to mRNA assays of cytokines.

**Bodipy staining and FACS**. The splenocytes and tumor tissue-derived cells were incubated with Fc Block (#553142, BD Bioscience) prior to staining with PerCP/Cy5.5 anti-CD45 (#103132, Biolegend) and anti-F4/80 antibodies (#45–4801, eBioscience) for 15 min at 4 °C, followed by two washes in Sorting Buffer. Cells were then resuspended in 500 µl of Bodipy 493/503 at 0.5 µg ml$^{-1}$ in PBS for 15 min. The sorting experiments were performed on FACSAriaII(BD, Bioscience). The Geometric mean fluorescence intensity (MFI) of Bodipy 493/503 in CD45+F4/80+ cells of each group was measured.

**Statistical analysis**. Statistical analyses were performed using GraphPad Prism (GraphPad Software, Inc.). The survival data were analyzed using a Gehan-Breslow-Wilcoxon test. All the other data were expressed as the means ± s.e.ms. and were analyzed using either one-way ANOVA or two-tailed unpaired Student's t-test. For each parameter of all data presented, * indicates $P < 0.05$, ** indicates $P < 0.01$, and *** indicates $P < 0.005$.

**Data availability**. The microarray data have been deposited in GEO under the accession code GSE80065. Other data that support the findings of this study are available within the article and its Supplementary information files or from the corresponding author upon reasonable request.

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

## Acknowledgements

This work was supported in part by award numbers 81672693 (H.M.) from the National Natural Science Foundation of China (NSFC), cstc2017jcyjBX0071 (H.M.) from the Foundation and Frontier Research Project of Chongqing, 31770931 (W.Y.) from NSFC, 2017A030306030 (W.Y.) from Guangdong Natural Science Funds for Distinguished Young Scholar and Youth 1000 Talent Plan (Y.L.).

## Author contributions

H.M., W.Y., Y.L., W.X., R.S., X.K., X.Z., P.C., L.Z., A.H., R.W., Y.Z., K.Z., Y.L., Y.M. and H.L. performed the experiments. H.M., H.L., W.Y. and Y.L. analyzed the data. S.S., J.Z., S.Y., F.H., L.G. and C.S. contributed to discussions. H.M. designed the project and wrote the manuscript. H.M. is the guarantor of this work, has full access to all the data and takes full responsibility for the integrity of the data.

## Additional information

**Competing interests:** The authors declare no competing interests.

