## [Peer Review File · Nature Communications]

Reviewers' comments:

Reviewer #1 (Remarks to the Author):

This is an interesting study addressing the importance of macrophage lipid metabolism during tumor progression. The authors found that tumor cells reprogram macrophage lipid metabolism inducing the accumulation of TAGs, DAGs and MGLL, which resulted in lipid overloading. MGLL suppress CB2/TLR4 signaling in macrophages, inhibiting CD8+ T cells response. While the study is of interest, there are some concerns that need to be addressed.

Specific points:

- 1) Figure 1. Additional gene expression analysis is required. What is the effect of tumor cells on transporters and enzymes that regulate FA uptake. The authors should measure LPL, ANGPTL4 and CD36 expression in TAM.
- 2) While MGLL expression appears to be inhibited in TAM, ATGL is increased. Why the authors observed an accumulation of TAGs and DAGs. Is there an increase in FA synthesis or MGLL re-acylation to TAGs.
- 3) FA oxidation has been shown to play an important role in macrophage activation. Is MGLL deficiency or overexpression affecting macrophage FA oxidation?. What is the effect of CB2 receptor in regulating FA oxidation.
- 4) Figure 4f. The authors need to include WB analysis for JNK and p-p65 and the quantification of the blot.
- 5) Figure 4h. There important control conditions missing in this experiment. The authors should include the effect igG IP in untreated and LPS-treated cells.

Reviewer #2 (Remarks to the Author):

This is an interesting study that provides new information regarding the changes in lipid metabolism of tumor-associated macrophages. The study is relevant, the experiment are generally well-performed, and the results are relevant.

Comments:

1. The literature reports a dichotomy between M1/M2 macrophages, yet this is likely an artificial oversimplification of many different macrophage phenotype (Xue et al, Immunity. 2014 Feb 20;40(2):274-88). I think it would be preferable to interpret the findings in light of this more complex reality of macrophage heterogeneity.
2. The authors report that the M1 protective phenotype was associated with production of more pro-inflammatory cytokine production such as IL-1b and TNF. However, a large body of literature supports the notion of cancer inflammation has deleterious effects, and recent data from the CANTOS trial demonstrates that anti-IL-1b therapy have beneficial effects against cancer. How can these observations be explained: do the authors think that the higher proinflammatory production of cytokines in the macrophages with transgene MGLL is protective or deleterious?
3. Lines 247-249: "the MGLL transgene-mediated tumor suppression (tumor volume and survival) was rescued by the addition of a CB2 transgene (Fig. 5b,c)". I guess the authors mean that tumor suppression was 'reversed' and not 'rescued'.
4. Did the authors check the reproducibility of the most important sets of results? No information was given if any data were repeated and found reproducible.
5. The information regarding the patients with colorectal cancer that were assessed with

immunohistochemistry is very scarce. What stadium was the disease, and what time during diseases was tissue histology done.

Reviewer #3 (Remarks to the Author):

Review of Monoacylglycerol lipase regulates cannabinoid receptor 2-dependent macrophage activation and cancer progression by Xiang/Shi/Kang/Zhang et al. for Nature Communications. Summary: This manuscript examines the reprogramming of TAMs in various cancer models (genetically engineered mouse models GEMMS) and xenografts (cell lines, not xenografts/PDX). Authors focused on monoacylglycerol lipase (MGLL) a lipase that regulates last step of TAG hydrolysis; it breaks down monoacylglycerols into FFA and glycerol backbone. Relevant to this paper is that it also hydrolyzes endocannabinoid 2-arachidonoylglycerol (2-AG) which signals through cannabinoid receptor 2 (CB2). Authors nicely use gain and loss of function, in vitro, in vivo models as well as fixed and fresh human samples to support findings. To review, authors first show that TAMs increase in several cell-line (not orthotopic) models and that they are lipid laden compared to splenic macs. This is likely due to downregulation of MGLL since this could be reversed with overexpression (a novel mouse model they made expressing myeloid MGLL (TGMGLL)). MGLL transgenic overexpression in myeloid cells decreased progression of both cell line CD38 model. This decrease in progression was lost in a RAG background suggesting a lymphoid involvement. Isolation of CD8 and CD4 cells showed MGLL affected primarily CD8 Tc cells not Cd4, which was supported by a loss of MGLL effect with anti-CD8 antibody in very nice convincing studies. CD38 mets to lung were not changed per mouse (they all got tumors, but there were fewer of them- this was presented in a confusing out of order way). There was a significant reduction in mets to liver detected in TGMGLL mice. A conditional MGLL myeloid KO was created (details on this were a little unclear) and tumors grew faster (the opposite of TGMGLL). TGMGLL TAMs and BMDM were more M1 like and less M2 like with the myeloid MGLL KO showing an opposite phenotype than the TG overexpresser (but not splenic macs). To get at mechanism, they show that TGMGLL decreased 2AG concentrations (the ligand for CB2) which led to increases in CB2 in several tumor types. CB2 increased over time in MC38 tumors too. Also tumor cells increased CB2 on peritoneal macs. Generation of a CB2 myeloid overexpresser did not change cytokines in splenic macs but did blunt the MGLL -induced increase in M1 like markers and blocked decreased of M2 markers in BMDMs and TAMs. The data show that 2AG blocks inflammatory pathways. CB2 colocalizes with TLR4 (in a 2AG dependent manner) which is a very important novel finding (using IHC and coIP). Further, using a genetic approach, CB2 mediated regulation of cytokines is blunted in TLR4KO TAMs. MGLL mediated regulation of T cell cytokines was ablated in myeloid CB2 transgenic mice. Tumor phenotypes regulated by MGLL were abrogated in CB2 mice, including liver mets. The mix of MC38 and macrophages in Fig 5g is an excellent study demonstrating the importance of the MGLL or CB2 macs (rather than other myeloid cells- a concern I note below). Antagonists to CB2 decreased tumor progression and improved survival in several tumor models (MC38, 4T1, Ct26) through myeloid cells. In very nice human findings, they show that MGLL is lower in tumors but not adjacent tissue. Critically, MGLL expression levels positively correlated with survival.

Overall, the manuscript is an excellent and well-done study using genetic and pharmacologic studies to study the MGLL/CB2/TLR4 pathway in TAMs. A few modifications to the style and explanations would add to the flow and clarity of the paper. In addition, the possibility of an additional experiment and possible in silico analysis would strengthen the paper. Finally, additional discussion of certain rationales for experiments and the role of CB2 antagonist in metabolic parameters is important since it was hard to follow in sections. Some overstatements and improper nomenclature (macrophage instead of myeloid) should be corrected as notes in minor and major comments. Of note, authors use several mac types and cancer cell lines, so additional notes in results or on figures will help reader follow along. Taken together, it is an excellent study with important clinical value. I believe that with minor editorial and/or additional analysis, this manuscript is worth of publication.

Major comments:

- 1) Are there variants or copy number changes in MGLL, CB2 associated with cancer (perhaps TCGA database cbiportal.org) which could strengthen this paper.
- 2) Since CB2 regulates appetite, are there metabolic consequences of antagonism or overexpression that could regulate tumor progression (glucose, leptin, adiponectin, etc.). A quick search found this paper for example <https://www.ncbi.nlm.nih.gov/pmc/articles/PMC4654496/>
- 3) Critically, if Fig 1c shows a survival benefit with tumors going out to day 25–40, why is the growth in Fig b only showing till day 10. The tumor growth should follow until mice drop out. Likewise, in Fig 2d the conditional KO for MGLL had tumors progressing faster out to day 10 but what about past that time? A Kaplan Meier survival plot like 2c for this model should be shown. Likewise, figure 2L and M. I do not understand why the tumor growth comparisons are at day 10 but survival goes out to later. If too many mice are dropping out and stats are NS, then give the rationale and just state that comparisons are made at day 10. This comment pertains to all tumor studies at day 10 whereas survival is out to > 30 days. This data should already exist and should be added to the paper, or justified why day 10 was used.
- 4) Using the TGMGLL model in Fig 2, authors show a delay in tumor volume in b. These error bars are quite large (they are SEMs, and only an N of 10) so the stats should be confirmed for significance.
- 5) Why were transwell migration assays conducted without serum. I do not think that the macrophages would be healthy- especially after 48 hours. Plus, this is a nonphysiologic approach. What was this rationale? I wonder if some of the migration assays would be different if serum were on board. This should be repeated with serum or justified in text.
- 6) The generation of the cKO MGLL mouse is unclear to me. How is this an inducible model? Is DTA diphtheria toxin- authors state that "DTA was used for negative selection"? I think that this is just a tissue specific model and not inducible. Furthermore, I understand the Cre recombination to remove exon 3 and LysM was used to drive Cre. This is a myeloid KO as mentioned in methods. However the nomenclature is indicated as "Mac-mgll-KO" where I assume Mac is macrophage. This is not correct since it is well known that LysM drives cre action in myeloid cells like neutrophils, monocytes, macrophages and likely other immune cells. Authors cite some old papers for the LysM Cre model references. See these 2 recent papers on LysM cre deletion of genes from other myeloid cells. Authors should annotate model as Myeloid mgll KO but note in discussion that immune cells other than macrophages may drive phenotype. Likewise, The CB2 overexpresser is CD11b driven, it is not a "macrophage-specific" model. Please correct. This is an important comment since the field has a lot of confusing nomenclature.
<https://www.ncbi.nlm.nih.gov/pmc/articles/PMC4105345/>
<https://www.frontiersin.org/articles/10.3389/fimmu.2017.01618/full>

Minor comments:

Typos throughout should be carefully fixed- Percific Blue – Pacific blue in fig 3. Santa Cruz, not Santa.

Abstract/ Introduction: It is interesting that through the title, abstract, and introduction, it is unclear what cancer(s) are being studied. This should be noted somewhere that this is primarily a colon cancer study with confirmation in other models.

Results:

- 1) P values are usually written after Fig reference in results to help judge significance of each statement.
- 2) Authors need to introduce the type of cell lines/cancer they are studying (Fig 1).
- 3) The flow in Fig 1a is unclear and should be labelled throughout for what each gate selects. This can be supplement. Define what is TSM (it is defined in supplement but not helpful for Fig 1).
- 4) Sup Fig 1 g is unclear what blue circle is. I assume mono culture but the bar for coculture next to legend is unclear.
- 5) I do not think that diagrams Fig 2a or e are necessary since this is pretty straightforward.

6) In suppl Fig 2, authors show that there is no defect in migration in coculture but I believe that these are peritoneal macs. It is unclear from the writing in results or legends what types of macs these are. Perhaps primary TAMs or another type of macrophage should be tested.

7) Please clarify this sentence- what is regulated (breakdown) "2-AG is regulated by the catabolic enzyme MGLL"

Methods:

1) "gene operated" is an unusual way to describe. Might change it to "transgenic" or genetically engineered mouse model (GEMM)"

2) spell out DTA. What is DTA doing here.

3) Incorrect terminology: A xenograft is a line across species like human into mouse.

4) A table of patient characteristics should be provided as supplement (age, tumor stage, BMI, smoker, drugs, etc if you have the information).

Discussion:

1) Subcutaneous models would be more valuable if they were injected orthotopically. This is especially true for breast cancer models where the fat pad of the mammary gland modifies growth. It should be noted in discussion that this is a caveat to interpretation.

2) The cartoon is nice but a little hard to read with the syringes there, etc.

Response to Reviewers' comments:

We thank the reviewers for their thoughtful and positive comments as well as the opportunity to respond. We have added new data to the paper and revised the paper to specifically address the concerns raised by you and the reviewers. We have marked the changes with red fonts. We also modified the formats according to the checklist provide by the journal. We believe we now have a more rigorous manuscript. We trust that the manuscript will now be acceptable for publication in Nature Communications.

Our responses to the reviewers are enumerated below:

Question 1: Reviewer #1 pointed out that “1) Figure 1. Additional gene expression analysis is required. What is the effect of tumor cells on transporters and enzymes that regulate FA uptake. The authors should measure LPL, ANGPTL4 and CD36 expression in TAM.”

Answer: We thank the reviewer for the suggestion. We have performed this experiment and demonstrated that mRNA levels of LPL and ANGPTL4 in macrophages were not regulated notably by tumor cells. However, the CD36 mRNA level was induced by the tumor cells modestly. We added those data in Supplementary Figure 1b. This result was also consistent with our microarray analysis, which was deposited in GEO under the accession codes GSE80065 and GSE80066.

Question 2: Reviewer #1 asked that “2) While MGLL expression appears to be inhibited in TAM, ATGL is increased. Why the authors observed an accumulation of TAGs and DAGs. Is there an increase in FA synthesis or MGLL re-acylation to TAGs. ”

Answer: We completely agree with this reviewer. The mRNA level of ATGL in TAM was not affected, while the expression of ABHD5, the activator of ATGL, was significantly induced (Supplementary Figure 1b), indicative an increase of ATGL activity (not validated). The mRNA level of fatty acid synthase (FASN) was also not altered in TAM versus TSM (Supplementary Figure 1b). Interestingly, the expression of key enzymes DGAT1/2, which produces TAGs from DAGs, was largely induced in TAM. Besides, the MGLL deficiency-induced MAG accumulation might activate MGATs (producing DAGs from MAGs) or DGATs (producing TAGs from DAGs). So, the lipid accumulation in TAM might result from multiple factors, in which MGLL was validated to be a very important contributor (Supplementary Figure 1d-g). However, the roles of other factors like DGAT1/2 in TAM lipid deposition need to be validated in future.

Question 3: Reviewer #1 pointed out that “3) FA oxidation has been shown to play an important role in macrophage activation. Is MGLL deficiency or overexpression affecting macrophage FA oxidation? What is the effect of CB2 receptor in. regulating FA oxidation.”

Answer: We thank the reviewer for raising this very insightful question! We examined the FA oxidation in peritoneal macrophages from the WT, Tg^{MGLL} (Myeloid transgene of MGLL) and Tg^{CB2} (Myeloid transgene of CB2) mice by measuring the mitochondrial oxygen consumption rates (OCR) with seahorse study. We didn't see any significant differences in OCR between those three kinds of macrophages (See the results below). This experiment indicated that both MGLL and CB2 did not affect macrophage fatty acid oxidation notably. To keep a good flow, we suggested not putting this data in the manuscript. Thank you!

Mitochondrial oxygen consumption rates (OCRs) of peritoneal macrophages from the WT ($n=3$), Tg^{MGLL} ($n=3$) and Tg^{CB2} ($n=3$) mice were measured under different treatments in 96-well plates by using a XF Cell Mito Stress Test Kit (#101706-100, Seahorse Bioscience, USA) on the XFe96 Extracellular Flux Analyzer (Seahorse Bioscience, USA) according to the Manufacturer's instruction and our previous study (Hongming Miao, et al., Cell Reports 2014). Basal cellular OCRs were recorded without metabolic inhibitors or uncouplers. ATP synthase was inhibited with 2 $\mu\text{g/ml}$ oligomycin, followed by uncoupling of the respiratory chain from oxidative phosphorylation by stepwise titration with 1 $\mu\text{g/ml}$ carbonyl cyanide *p*-(trifluoromethoxy) phenylhydrazone (FCCP) to achieve maximal OCRs. Finally, rotenone (Mito Inhibitor B), a Complex I inhibitor (1 $\mu\text{g/ml}$), and antimycin A (Mito Inhibitor A), a Complex III inhibitor (1 $\mu\text{g/ml}$) were combined to totally inhibit the mitochondrial respiratory chain. The results were presented as pmoles/min OCRs.

Question 4: Reviewer #1 pointed that “4) Figure 4f. The authors need to include WB analysis for JNK and p-p65 and the quantification of the blot.”

Answer: We thank the reviewer for the suggestion. We have added values under the bands to indicate the relative expression according to the density (See in Figure 4f).

Question 5: Reviewer #1 pointed out that “5) Figure 4h. There important control conditions missing in this experiment. The authors should include the effect igG IP in untreated and LPS-treated cells.”

Answer: Thanks for the suggestion. We totally agree with the reviewer. We now have added the IgG-immunoprecipitated group (See in Figure 4h).

Question 6: Reviewer #2 suggested that “1. The literature reports a dichotomy between M1/M2 macrophages, yet this is likely an artificial oversimplification of many different macrophage phenotype (Xue et al, Immunity. 2014 Feb 20;40(2):274-88). I think it would be preferable to interpret the findings in light of this more complex reality of macrophage heterogeneity.”

Answer: We totally agree with the reviewer. We thank the reviewer for this suggestion that certainly strengthens the paper! We have now modified the wording regarding to M1/M2 throughout the manuscript. We emphasize that M1 or M2 is just an ideal state in vitro, while the tissue macrophages are always heterogeneous. So, we described the macrophages using the words like "M1-like", "M2-like", "proinflammatory" or "anti-inflammatory". We think these descriptions are more appropriate and easy to understand. Those changes were marked red in the text.

Question 7: Reviewer #2 pointed out that “2. The authors report that the M1 protective phenotype was associated with production of more pro-inflammatory cytokine production such as IL-1b and TNF. However, a large body of literature supports the notion of cancer inflammation has deleterious effects, and recent data from the CANTOS trial demonstrates that anti-IL-1b therapy have beneficial effects against cancer. How can these observations be explained: do the authors think that the higher proinflammatory production of cytokines in the macrophages with transgene MGLL is protective or deleterious?”

Answer: The protective effect of M1-like macrophages in cancer comes from the increase of proinflammatory cytokines (TNF α , etc) and reduction of anti-inflammatory cytokines (IL-10, etc). Generally, M1-like macrophages mediate Th1 and anti-tumor responses, while M2-like macrophages do the opposite. Just like a double-edged sword, inflammation has a pro-tumor effect in cancer cells and an anti-tumor function in tumor immunology. Undoubtedly, MGLL promoted M1 cytokine production and simultaneously attenuated M2 cytokine expression in tumor associated macrophages. Therefore, we believe "the macrophages with transgene MGLL is protective". However, whether the comprehensive benefits (activation of CD8+ T cells and tumor inhibition) came from a single or multiple cytokines needs further to be validated.

As mentioned by the reviewer, CANTOS trial (Ridker PM, et al., Lancet 2017) demonstrated that IL-1 β inhibition with canakinumab could reduce incident lung cancer and lung cancer mortality. Actually, our study is totally different from that trial in multiple aspects, such as research object (Patients with cancer and atherosclerosis vs mice with only cancer), cancer type (lung cancer vs colorectal cancer) and treatment (systemic IL-1 β inhibition vs myeloid cell-specific intervention of MGLL or CB2). Whether macrophage MGLL/IL-1 β axis would regulate the development of colorectal cancer or lung cancer needs further to be investigated. The authors in CANTOS trial also claimed that "Our hypothesis-generating data suggest the possibility that anti-inflammatory therapy with canakinumab targeting the interleukin-1 β innate immunity pathway could significantly reduce incident lung cancer and lung cancer mortality. Replication of these data in formal settings of cancer screening and treatment is required. "

Question 8: Reviewer #2 pointed out that "3. Lines 247-249: "the MGLL transgene-mediated tumor suppression (tumor volume and survival) was rescued by the addition of a CB2 transgene (Fig. 5b,c)". I guess the authors mean that tumor suppression was 'reversed' and not 'rescued'."

Answer: We thank the reviewer for the careful review. We totally agreed with that and replaced "rescued" with "reversed".

Question 9: Reviewer #2 pointed out that "4. Did the authors check the reproducibility of the most important sets of results? No information was given if any data were repeated and found reproducible."

Answer: We thank the reviewer for the notice. We did repeat the most important results. We have now added this information in the Figure legends. The changes were marked red (See in legends of Figure 1c, 2a, 2e, 3a, 4g, 4h and 5b).

Question 10: Reviewer #2 pointed out that “5. The information regarding the patients with colorectal cancer that were assessed with immunohistochemistry is very scarce. What stadium was the disease, and what time during diseases was tissue histology done.”

Answer: We apologize for the confusion to the reviewer. We have added more clinical information about these CRC patients in the method section of “Immunohistochemistry of patient samples”. Thanks.

Question 11: Reviewer #3 pointed out that “1) Are there variants or copy number changes in MGLL, CB2 associated with cancer (perhaps TCGA database cbiportal.org) which could strengthen this paper.”

Answer: We thank the reviewer for the thoughtful suggestion. It should be pointed out that we couldn’t get any gene expression information in tumor associated macrophages in a special type of tumor in the TCGA or other public databases. Even so, we still got some interesting information about MGLL and CB2 in cancers in two public databases. Given the reality that we are mainly investigating the roles of MGLL and CB2 in tumor associated macrophages, but not cancer cells, we suggest not including those data in the revised manuscript. Thanks.

(1) We searched the TCGA database provided by the reviewer and found that MGLL in colorectal cancer tissues displayed mainly in the forms of mutation or deep deletion (See the data below in A). However, we didn’t obtain any information about CB2 in colorectal cancer (See the data below in B).

(2) We also searched the expression of MGLL and CB2 associated colon cancer in another database (<http://merav.wi.mit.edu/SearchByGenes.html>). We found that MGLL expression in primary tumors was largely decreased as compared to the normal colon tissues. However, we didn’t see a notably difference in the CB2 (encoded by *cnr2*) expression between the primary tumors and normal tissues (See the data below in C).

Question 12: Reviewer #3 pointed out that “2) Since CB2 regulates appetite, are there metabolic consequences of antagonism or overexpression that could regulate tumor progression (glucose, leptin, adiponectin, etc.). A quick search found this paper for example <https://www.ncbi.nlm.nih.gov/pmc/articles/PMC4654496/>”

Answer: We thank the reviewer for the thoughtful idea involving metabolic consequences as mechanisms for CB2-mediated tumor progression. Actually, CB2 was conditionally engineered in myeloid cells in our present study. We didn't find any significant differences in food intake (See the data below in A), blood glucose levels (See the data below in B) or blood leptin levels (See the data below in C) between the wild type and myeloid CB2 transgenic mice. Therefore, we concluded that metabolic consequences (glucose, leptin, etc.) might not be a major regulator of tumor progression in our experimental system. However, if the CB2 signaling was intervened systemically (such as treatment with agonists or antagonists), but not myeloid-specifically, the phenotype of metabolic consequences might be obvious (such as the study mentioned by the reviewer). In this case, the tumor progression might be influenced.

intake of male WT and Tg^{CB2} mice were measured weekly from 7-week to 8-week of age (n=5). (B) Fed blood glucose levels of 8-week old male mice were measured (n=5). (C) Fed

blood leptin levels of 8-week old male mice were measured with a Mouse/Rat Leptin Quantikine ELISA kit (#MOB00,R&D Systems).(n=5)

Question 13: Reviewer #3 pointed out that “3) Critically, if Fig 1c shows a survival benefit with tumors going out to day 25—40, why is the growth in Fig b only showing till day 10. The tumor growth should follow until mice drop out. Likewise, in Fig 2d the conditional KO for MGLL had tumors progressing faster out to day 10 but what about past that time? A Kaplan Meier survival plot like 2c for this model should be shown. Likewise, figure 2L and M. I do not understand why the tumor growth comparisons are at day 10 but survival goes out to later. If too many mice are dropping out and stats are NS, then give the rationale and just state that comparisons are made at day 10. This comment pertains to all tumor studies at day 10 whereas survival is out to > 30 days. This data should already exist and should be added to the paper, or justified why day 10 was used.”

Answer: Thanks for the suggestions. We have added the survival data for the conditional MGLL-KO mice bearing MC-38 tumors (See in Figure 2d).

With regard to the time window of tumor observation in the present study, we did design to measure tumor volumes on day 7, day 10, day 14 and day 21 after inoculation in previous experiment. However, we found that the data obtained after day 14 were not accurate and sensitive, because those large tumors were usually accompanied with necrosis and escharosis (resulting in irregular shape). All the inoculated tumors will have no obvious difference in volume over time. We don't think the tumor volume in the later period of tumor-bearing mice can reflect the actual severity of tumor progression. Therefore, we just displayed the measuring data till day 10, which can be applied to all the study under all the treatment. We thank the reviewer again for suggestion.

Question 14: Reviewer #3 pointed out that “4) Using the TGMGLL model in Fig 2, authors show a delay in tumor volume in b. These error bars are quite large (they are SEMs, and only an N of 10) so the stats should be confirmed for significance. ”

Answer: Thank the reviewer for the careful review. We rechecked the raw data and found we had mistaken stand deviation (STDEV) for SEM [=STDEV/SQRT(10)]. We now have corrected the SEM data and the error bars. We confirmed that the difference is significant (See in Figure 2a).

Question 15: Reviewer #3 pointed out that “5) Why were transwell migration assays conducted without serum. I do not think that the macrophages would be healthy- especially after 48 hours. Plus, this is a nonphysiologic approach. What was this rationale? I wonder if

some of the migration assays would be different if serum were on board. This should be repeated with serum or justified in text.”

Answer: We apologize for the confusion to the reviewer. The upper chamber maintained the colorectal cancer cells MC-38 (**not macrophages**) without serum, while the lower chamber contained enough conditioned medium (with 5% FBS) from WT or Tg^{MGLL} macrophages. We didn't introduce the method for the preparation of conditioned medium in the first edition of manuscript. Thanks to the reviewer for the carefulness. We have now added “preparation of conditioned medium” in the method section. We also explained why we chose serum free medium in upper chambers in the section of “Transwell assays” in Methods. We found that 5% FBS in upper chambers could largely potentiate cell proliferation and migration, which made it difficult to choose the observing time point and calculate the migrated cells. As our experience, the serum concentration in upper chambers should be modified according to different cancer cells and different observing time points. Our experimental condition of the present study was tested and optimized.

Question 16: Reviewer #3 pointed out that “6) The generation of the cKO MGLL mouse is unclear to me. How is this an inducible model? Is DTA diphtheria toxin- authors state that “DTA was used for negative selection”? I think that this is just a tissue specific model and not inducible. Furthermore, I understand the Cre recombination to remove exon 3 and LysM was used to drive Cre. This is a myeloid KO as mentioned in methods. However the nomenclature is indicated as “Mac-mgll-KO” where I assume Mac is macrophage. This is not correct since it is well known that LysM drives cre action in myeloid cells like neutrophils, monocytes, macrophages and likely other immune cells. Authors cite some old papers for the LysM Cre model references. See these 2 recent papers on LysM cre deletion of genes from other myeloid cells. Authors should annotate model as Myeloid mgll KO but note in discussion that immune cells other than macrophages may drive phenotype. Likewise, The CB2 overexpresser is CD11b driven, it is not a “macrophage-specific” model. Please correct. This is an important comment since the field has a lot of confusing nomenclature.

<https://www.ncbi.nlm.nih.gov/pmc/articles/PMC4105345/>

<https://www.frontiersin.org/articles/10.3389/fimmu.2017.01618/full>”

Answer: We apologize to the reviewer for the confusion. The full name of DTA is Diphtheria toxin A chain. We have modified it in the text. As the reviewer mentioned, the cKO MGLL mouse is just a tissue specific model and not inducible. We totally agree with the reviewer. The LysM-driven Cre just makes a myeloid KO. Therefore, we have changed “Mac-mgll-KO” as “Myeloid-mgll-KO” throughout the manuscript. We also corrected other inappropriate wording like “macrophage-specific” completely. Besides, we also discussed the potential importance of other myeloid cells other than macrophages in regulating colorectal cancer progression in the section of Results (See in page 11, line 261-263).

Question 17: Reviewer #3 pointed out that “Minor comments: Typos throughout should be carefully fixed- Percific Blue – Pacific blue in fig 3. Santa Cruz, not Santa.”

Answer: We really appreciate the reviewer for pointing out those errors! We have now corrected throughout the manuscript.

Question 18: Reviewer #3 pointed out that “Minor comments: Abstract/ Introduction: It is interesting that through the title, abstract, and introduction, it is unclear what cancer(s) are being studied. This should be noted somewhere that this is primarily a colon cancer study with confirmation in other models.”

Answer: We totally agree with the viewer. We have noted it in the revised abstract as "With primary investigation in colon cancer and confirmation in other cancer models, here we determined"(Page 2, line 30-31). Thanks for the suggestion.

Question 19: Reviewer #3 pointed out that “Minor comments: Results:

- 1) P values are usually written after Fig reference in results to help judge significance of each statement.
- 2) Authors need to introduce the type of cell lines/cancer they are studying (Fig 1).
- 3) The flow in Fig 1a is unclear and should be labelled throughout for what each gate selects. This can be supplemented. Define what is TSM (it is defined in supplement but not helpful for Fig 1).
- 4) Sup Fig 1 g is unclear what blue circle is. I assume mono culture but the bar for coculture next to legend is unclear.
- 5) I do not think that diagrams Fig 2a or e are necessary since this is pretty straightforward.
- 6) In suppl Fig 2, authors show that there is no defect in migration in coculture but I believe that these are peritoneal macs. It is unclear from the writing in results or legends what types of macs these are. Perhaps primary TAMs or another type of macrophage should be tested.
- 7) Please clarify this sentence- what is regulated (breakdown) “2-AG is regulated by the catabolic enzyme MGLL” ”

Answer: We really appreciate the reviewer for the carefulness. Our responses are enumerated below:

(1) We tried our best to state the significance using the word, such as "notably", "Largely" and "significantly". If the reviewer or the editorial insists on displaying the P values in the "Results" section, we will do it. Thanks a lot.

(2) We have introduced the type of cell lines in the first paragraph of "Results" section.

(3) We have modified the labels of flow in Fig 1a. We also defined TSM and TAMs in the legend of Figure 1c.

(4) We apologize for the confusion to the reviewer. We made a mistake in labeling. Actually, the blue cycle indicates the mono culture as the reviewer speculated. We have now corrected it.

(5) We agree with the reviewer and have discarded those two diagrams.

(6) We apologize for the confusion to the reviewer. In supplementary Figure 2d-2e, what we tested was colon cancer cell MC-38, but **not macrophages**. We investigated the role of macrophage-derived factors on MC-38 migration. There is no rationale for detecting the migration of macrophages in the present study. Thanks.

(7) Thanks for the suggestion. We have replaced "regulated " with "broke down".

Question 20: Reviewer #3 pointed out that "Minor comments: Methods:

1) "gene operated" is an unusual way to describe. Might change it to "transgenic" or genetically engineered mouse model (GEMM)"

2) spell out DTA. What is DTA doing here.

3) Incorrect terminology: A xenograft is a line across species like human into mouse.

4) A table of patient characteristics should be provided as supplement (age, tumor stage, BMI, smoker, drugs, etc if you have the information). "

Answer: Thanks for the suggestions. Our responses are listed below:

(1) We totally agree with the reviewer. We have corrected it throughout the manuscript.

(2) The full name of DTA is Diphtheria toxin A chain. We have put the full name in the text. The DTA is used for negative selection. It means that the DTA-resistant cells were not wanted.

(3) We have modified the wording throughout the manuscript. The word "xenograft" was replaced by "tumor", "inoculation" or "injection".

(4) We have made a table of patient characteristics as supplement (See in Supplementary Table 1). Thanks.

Question 21: Reviewer #3 pointed out that “Minor comments: Discussion:

1) Subcutaneous models would be more valuable if they were injected orthotopically. This is especially true for breast cancer models where the fat pad of the mammary gland modifies growth. It should be noted in discussion that this is a caveat to interpretation.

2) The cartoon is nice but a little hard to read with the syringes there, etc. ”

Answer: Thanks for the suggestion. (1) We have added this notion in "Subcutaneous tumor models" in Method section. (2) We have deleted the syringe signs to avoid confusion in the cartoon.

Reviewers' comments:

Reviewer #1 (Remarks to the Author):

The authors have addressed most of my concerns. As mentioned in my original repose, the authors should include the WB analysis for TOTAL JNK and normalize the JNK activation (phosphorylation) to this value. Same thing for p65 (missing) and p-p65.

Reviewer #2 (Remarks to the Author):

The authors responded appropriately to my suggestions.

Reviewer #3 (Remarks to the Author):

They have carefully addressed my concerns as well are concerns of Rev 1 and 2. I agree with their additions, deletions, and edits. I do not insist on them adding P values in results- it is fine as is. Accept.

Response to Reviewers' comments:

We thank the reviewers for their efforts and positive comments on our paper entitled "**Monoacylglycerol lipase regulates cannabinoid receptor 2-dependent macrophage activation and cancer progression**". We revised the paper to specifically address the concerns raised by the reviewers. We have marked the changes with red fonts. We believe we now have a more rigorous manuscript which is acceptable for publication in Nature Communications.

Our responses to the reviewers are enumerated below:

Question 1: Reviewer #1 pointed out that "The authors have addressed most of my concerns. As mentioned in my original repose, the authors should include the WB analysis for TOTAL JNK and normalize the JNK activation (phosphorylation) to this value. Same thing for p65 (missing) and p-p65."

Answer: We thank the reviewer for the suggestion. We apologize for our misunderstanding in the first revision. We now have added the analysis of total JNK and p65 in the experiment in Figure 4f. The primary conclusion is not affected by the new analysis.

Question 2: Reviewer #2 responded that "The authors responded appropriately to my suggestions."

Answer: We thank the reviewer for the positive comments.

Question 3: Reviewer #3 pointed out that "They have carefully addressed my concerns as well are concerns of Rev 1 and 2. I agree with their additions, deletions, and edits. I do not insist on them adding P values in results- it is fine as is. accept"

Answer: We thank the expert again for the careful review and positive response.